# Misfolded polypeptides are selectively recognized and transported toward aggresomes by a CED complex

Joori Park[1,2,*], Yeonkyoung Park[1,2,*], Incheol Ryu[1,2], Mi-Hyun Choi[2], Hyo Jin Lee[2], Nara Oh[1,2], Kyutae Kim[2,3], Kyoung Mi Kim[2,†], Junho Choe[2,†], Cheolju Lee[3], Ja-Hyun Baik[2] & Yoon Ki Kim[1,2]

Misfolded polypeptides are rapidly cleared from cells via the ubiquitin–proteasome system (UPS). However, when the UPS is impaired, misfolded polypeptides form small cytoplasmic aggregates, which are sequestered into an aggresome and ultimately degraded by aggrephagy. Despite the relevance of the aggresome to neurodegenerative proteinopathies, the molecular mechanisms underlying aggresome formation remain unclear. Here we show that the CTIF–eEF1A1–DCTN1 (CED) complex functions in the surveillance of either pre-existing or newly synthesized polypeptides by linking two molecular events: selective recognition and aggresomal targeting of misfolded polypeptides. These events are accompanied by CTIF sequestration into the aggresome, preventing the additional synthesis of misfolded polypeptides from mRNAs bound by nuclear cap-binding complex. These events render cells more resistant to apoptosis induced by proteotoxic stresses. Collectively, our data provide compelling evidence for a previously unappreciated protein surveillance pathway and a regulatory gene expression network for coping with misfolded polypeptides.

[1] Creative Research Initiatives Center for Molecular Biology of Translation, Korea University, Seoul 02841, Republic of Korea. [2] Division of Life Sciences, Korea University, Seoul 02841, Republic of Korea. [3] BRI, Korea Institute of Science and Technology, Hwarangno 14-gil 5, Seongbuk-gu, Seoul 02792, Republic of Korea. * These authors contributed equally to this work. † Present addresses: Laboratory of Genetics and Genomics, National Institute on Aging, Intramural Research Program, National Institutes of Health, Baltimore, Maryland 21224, USA (K.M.K.); Stem Cell Program, Boston Children's Hospital, Massachusetts 02115, USA (J.C.). Correspondence and requests for materials should be addressed to Y.K.K. (email: yk-kim@korea.ac.kr).

The quantity and quality control of eukaryotic gene expression is coordinated at multiple steps[1]. After being processed in the nucleus, newly synthesized mRNAs are exported to the cytoplasm with their 5′-cap bound to the nuclear cap-binding protein complex (CBC), a heterodimer of cap-binding protein 80 (CBP80, also known as nuclear cap-binding protein 1 (NCBP1)) and either CBP20 (also known as NCBP2) or NCBP3 (refs 2,3). In the cytoplasm, CBC is replaced by the cytoplasmic eukaryotic translation initiation factor 4E (eIF4E) in a translation-independent manner[4]. Both CBC and eIF4E can recruit ribosomes[5]. CBC recruits ribosomes through an interaction with CBC-dependent translation (CT) initiation factor (CTIF), driving the first (or pioneer) round of translation[2,5–9]. On the other hand, eIF4E-dependent translation (ET) uses eIF4G to recruit ribosomes and enable protein synthesis in bulk[10].

In mammalian cells, misfolded polypeptides may arise owing to various intrinsic and extrinsic factors, including genetic mutations, abnormal translation generating defective ribosomal products (DRiPs), misfolding or aberrant modifications during or after translation, failure of ribosome quality control and environmental stresses[11–16]. Furthermore, truncated misfolded polypeptides may be synthesized as byproducts of translation-coupled mRNA surveillance pathways[17–20], such as nonsense-mediated mRNA decay, because this process necessitates at least a single round of translation to survey the existence of a premature termination codon along newly synthesized mRNAs[17,21].

To cope with misfolded polypeptides, eukaryotic cells have evolved highly sophisticated molecular mechanisms at several levels[12–16]. Molecular chaperones such as heat shock proteins help misfolded polypeptides refold into the appropriate three-dimensional conformation. Alternatively, misfolded polypeptides are degraded by the ubiquitin–proteasome system (UPS), preventing their accumulation within cells. However, when these two initial processes are overwhelmed or impaired, the misfolded polypeptides are prone to forming small cytoplasmic aggregates[22–24]. Aggregates of misfolded polypeptides are actively transported into a perinuclear structure, the aggresome, via dynein-mediated retrograde transport to the microtubule-organizing centre (MTOC). The aggresome is then surrounded by the intermediate filament, vimentin and eventually cleared by aggrephagy, a selective autophagic clearance process[14,25]. Aggresome clearance by aggrephagy may play a cytoprotective role in response to the accumulation of aggregates containing misfolded polypeptides when chaperones and the UPS are impaired[26]. In support of this, several previous reports have revealed that aggresome formation correlates with cell survival[27–29].

Histone deacetylase 6 (HDAC6) is involved in the selective recognition and movement of small cytoplasmic aggregates containing misfolded polypeptides towards the aggresome[27,30,31]. HDAC6 binds to unanchored ubiquitin C-terminal tails in the misfolded polypeptides through its C-terminal binder of ubiquitin zinc-finger/zinc-finger-ubiquitin-specific processing protease (ZnF-UBP) domain, and associates with dynein motors through its N-terminal dynein motor-binding domain[27,32]. Through these interactions, HDAC6 functions as a molecular adaptor to link misfolded polypeptide-containing small cytoplasmic aggregates to dynein motors, triggering the efficient movement of the aggregates to the aggresome.

Here we provide molecular and functional evidence that explains how either pre-existing or newly synthesized misfolded polypeptides are selectively recognized and form an aggresome that is eventually eliminated by aggrephagy. We found that CTIF, a scaffold protein in CT, is localized to misfolded polypeptide-containing aggresomes. Mechanistically, CTIF associates with both the eukaryotic translation elongation factor 1 alpha 1 (eEF1A1) and dynactin 1 (DCTN1) through its N-terminal region, forming a functional complex called CTIF-eEF1A1-DCTN1 (CED) complex. When the UPS is impaired, misfolded polypeptides form small cytoplasmic aggregates. Under these conditions, CTIF functions as a molecular adaptor that physically links the small cytoplasmic misfolded polypeptide-containing aggregates, which are selectively recognized by eEF1A1, to dynein motors via DCTN1. These small cytoplasmic aggregates are transported towards the aggresome via dynein-mediated retrograde movement. During this process, CTIF is also sequestered into the aggresome with the misfolded polypeptides. As a consequence, the efficiency of CT, which requires CTIF, is reduced, preventing further expression of misfolded polypeptides synthesized from mRNAs bound by CBC. Furthermore, down-regulation of each CED component renders cells more vulnerable to apoptosis induced by an accumulation of misfolded polypeptides. Collectively, these data provide molecular insight into a previously unappreciated protein surveillance pathway by which the cells cope with misfolded polypeptides.

## Results

**CTIF is localized in the aggresomes**. Our previous report revealed that endogenous CTIF is localized to the cytoplasmic side of the perinuclear region, supporting our model that CTIF is loaded onto the 5′-end of newly synthesized mRNAs that are being exported from the nucleus to the cytoplasm through the nuclear pore complex[6]. In our previous report, as well as in this study, we also observed that, in addition to the perinuclear region, endogenous CTIF was concentrated in one or two punctate cytoplasmic bodies that were closely juxtaposed to the outer membrane of the nuclear envelope (Supplementary Fig. 1). The endogenous CTIF-enriched cytoplasmic bodies overlapped with γ-tubulin (Fig. 1a), a component of MTOC[33], but not with DCP1A-enriched processing bodies (Supplementary Fig. 2a) where mRNA-degrading enzymes are enriched[34,35]. MTOC is the cellular site of aggresome formation, where small cytoplasmic aggregates of misfolded polypeptides are sequestered and eventually degraded by aggrephagy[22–24]. Therefore, using various approaches, we determined whether the CTIF-enriched cytoplasmic bodies are aggresomes.

Inhibition of the proteasome or autophagy is known to result in the accumulation of misfolded polypeptides, which can lead to aggresome formation[22–24]. The treatment of HeLa cells with the proteasome inhibitor MG132 increased the size of endogenous CTIF bodies (Fig. 1a,b) without affecting the number of CTIF bodies per cell (Fig. 1a,c). Furthermore, treatment with the autophagy inhibitor bafilomycin A1 markedly increased the size of endogenous and FLAG-tagged CTIF bodies (Fig. 1d and Supplementary Fig. 2b). Misfolded polypeptide aggregates are selectively moved towards the aggresomes via dynein- and microtubule-mediated retrograde transport[22–24]. In support that CTIF bodies are aggresomes, treatment with the microtubule-disrupting drug nocodazole markedly inhibited the formation of endogenous and FLAG-tagged CTIF bodies (Fig. 1d and Supplementary Fig. 2b).

Using aggresome marker proteins, we further determined whether these CTIF bodies are aggresomes. Endogenous or FLAG-tagged CTIF bodies significantly overlapped with previously known aggresome components following MG132 treatment (Fig. 1e and Supplementary Fig. 3a,b): (i) endogenous vimentin[22]; (ii) HDAC6 (ref. 27); and (iii) ubiquitin, the localization of which was determined by monomeric red fluorescent protein-tagged ubiquitin (mRFP-Ub)[36]. In addition, CTIF bodies overlapped with previously known aggresome-targeted misfolded polypeptides following treatment with MG132: (i) polypeptidyl-puromycin (polypeptidyl-puro; Supplementary Fig. 3c), which is a prematurely terminated translation product by

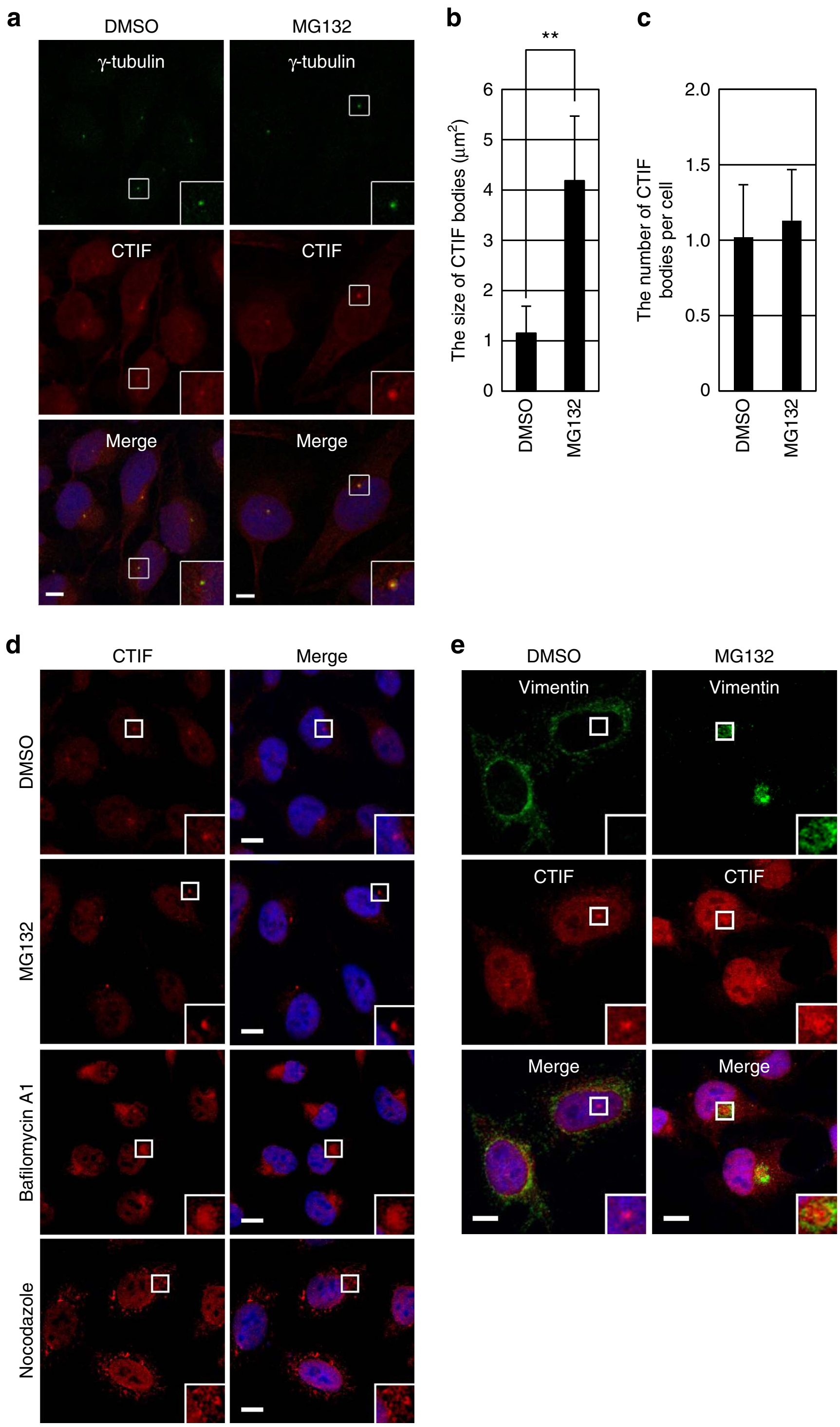


**Figure 1 | CTIF is localized to aggresomes.** (**a**) Immunostaining for endogenous CTIF (red) and γ-tubulin (green) in HeLa cells in the presence of either DMSO or MG132. (**b**,**c**) Quantification of the change in the size (**b**) and number (**c**) of CTIF-containing cytoplasmic bodies. Immunostained images of the cells in **a** were quantitated. The columns and error bars represent the mean and s.d. of three biological replicates. Two-tailed, equal-sample variance Student's *t*-tests were used to calculate the *P* values. $^{**}P<0.01$. (**d**) Immunostaining for endogenous CTIF (red) in HeLa cells treated with the indicated inhibitors for 12 h. (**e**) Immunostaining for endogenous vimentin (green) and CTIF (red) in HeLa cells. Nuclei were stained with DAPI (blue). Enlarged images of the boxed areas are shown in the lower-right corner of each image. All results are representative of three independent biological replicates ($n=3$). Scale bar, 10 μm.

puromycin treatment and, therefore, corresponds to DRiPs[37,38]; (ii) a green fluorescent protein (GFP)-fused cystic fibrosis transmembrane conductance regulator (CFTR)-ΔF508 (CFTR-ΔF508; Supplementary Fig. 3d), which is a chimeric protein composed of GFP and a mutant variant of CFTR harbouring a single amino-acid (phenylalanine) deletion at position 508 (refs 22,27); and (iii) synphilin 1-GFP (SYN1-GFP; Supplementary Fig. 3e), which associates with α-synuclein and

is targeted to cytosolic inclusion bodies[38–40]. CTIF bodies were also co-localized with another type of aggresome-targeted misfolded protein, GFP-250 (Supplementary Fig. 3f), which is a fusion protein composed of GFP and a 250-amino-acid fragment of a membrane transport factor p115, and forms aggresomes independently of MG132 treatment[41]. All these data clearly indicate that the observed CTIF bodies are aggresomes.

**CTIF is complexed with DCTN1 and eEF1A1**. To understand the molecular mechanism underlying CTIF localization to the aggresomes, we performed liquid chromatography-tandem mass spectrometry (LC-MS/MS) using an immunoprecipitate of FLAG-CTIF, and identified two cellular factors as CTIF-interacting proteins (Fig. 2a): DCTN1, the largest subunit of dynactin, which interacts with an intermediate chain of dynein motor protein and microtubule, and is involved in the retrograde movement of cargoes along microtubules[42]; and eEF1A1, an alpha subunit of the eEF1 complex and a GTP-binding protein responsible for the delivery of aminoacyl-tRNAs to the A site of a ribosome[43]. Interestingly, eEF1A1 is also known to directly bind both pre-existing and newly synthesized misfolded polypeptides during and after translation, and could generate a signal for aggresome formation[36,44,45]. The direct interactions of CTIF with DCTN1 and eEF1A1 were demonstrated by glutathione *S*-transferase (GST) pull-down experiments using recombinant proteins (Supplementary Fig. 4a,b). Moreover, the confocal microscopy results showed that DCTN1 and eEF1A1 overlapped with the CTIF aggresome upon MG132 treatment (Supplementary Fig. 4c). These data indicate that DCTN1 and eEF1A1 directly interact with CTIF and become enriched in aggresomes with CTIF when the UPS is impaired.

**CED complex shifts from the CT complex to aggresomes**. CTIF was originally identified as a CT component[6]. In this study, we showed that CTIF is localized to the aggresome (Fig. 1) and interacts with DCTN1 and eEF1A1 (Fig. 2a). We therefore investigated the possible interplay between the CTIF-containing CT complex and the aggresome. The results of immunoprecipitations (IPs) using cell extracts treated with RNase A showed that CT-specific factors (CBP80 and eIF4AIII) and common factors for CT and ET (eIF3b, a component of the eIF3 complex, and ribosomal protein S3 (rpS3), a component of the small subunit of the ribosome), but not an ET-specific factor (eIF4E), were preferentially enriched in the IPs of either endogenous CTIF (Fig. 2b) or FLAG-CTIF (Supplementary Fig. 5a). Interestingly, the observed enrichment of the tested proteins was significantly abolished when the cells were treated with MG132. In contrast, DCTN1 and eEF1A1 enriched in the IPs were unaffected by treatment with MG132. Taken together with the confocal microscopy results (Supplementary Fig. 4c), these data indicate that, whereas CTIF, DCTN1 and eEF1A1 associate with the CT complex under normal conditions, they dissociate from the CT complex and move towards the aggresome when the UPS is impaired. In agreement with this conclusion, CTIF and DCTN1 co-immunopurified with FLAG-eEF1A1 in a way that was unaffected by treatment with MG132 (Supplementary Fig. 5b). In contrast, a CT-specific factor (CBP80) and a common factor for CT and ET (rpS3), but not ET-specific factors (eIF4E and eIF4GI), preferentially co-immunopurified with FLAG-eEF1A1 in a MG132-dependent manner. Notably, DCTN1 and eEF1A1 were very weakly co-immunopurified with eIF4GI, an ET-specific factor (Supplementary Fig. 5c), indicating the specific association of CTIF, DCTN1 and eEF1A1 with the CT complex under normal conditions, and selective dissociation of these components from the CT complex upon UPS impairment.

It should be noted that the reduced association of CT components (CBP80 and eIF3b) with the CTIF complex upon MG132 treatment was significantly reversed by downregulation of either DCTN1 or eEF1A1 using specific small interfering RNAs (siRNAs; Fig. 2c). Furthermore, the downregulation of DCTN1 and eEF1A1 abolished the interactions of CTIF with eEF1A1 and DCTN1, respectively. These data indicate that DCTN1 and eEF1A1 promote the release of CTIF from the CT complex upon UPS impairment, and are necessary for the integrity of the CTIF-eEF1A1-DCTN1 complex, hereafter referred to as the CED complex. In the CED complex, CTIF functions as an adaptor protein to link eEF1A1 and DCTN1, which was evident from the IPs showing that downregulation of CTIF inhibited an interaction between eEF1A1 and DCTN1 (Supplementary Fig. 5d,e).

CED complex dissociation from the CT complex upon MG132 treatment implies that the CED complex may be released from mRNAs when the UPS is impaired. Indeed, the results of endogenous CTIF IP showed that MG132 treatment triggered the dissociation of CFTR-ΔF508 mRNA from CTIF (Fig. 2d). On the other hand, fluorescent *in situ* hybridization revealed that MG132 treatment did not significantly affect the intracellular distribution of CFTR-ΔF508 mRNA, although CFTR-ΔF508 protein accumulated in an aggresome under the same conditions (Fig. 2e). Therefore, we conclude that when the UPS is impaired, the CED complex dissociates from mRNAs and moves towards the aggresome without affecting the intracellular distribution of the mRNAs.

**CED-driven transport of misfolded polypeptides to aggresome**. Considering that (i) eEF1A1 binds to both pre-existing and newly synthesized misfolded polypeptides, and generates a signal for aggresome formation[36,44,45]; and (ii) that DCTN1 mediates the movement of cargoes along microtubules[42], the CED complex may play an active role in the aggresome formation of misfolded polypeptides. We examined this possibility using siRNAs against each CED component and four different types of misfolded polypeptides: polypeptidyl-puro (one of the DRiPs[37,38]; Fig. 3a,b); CFTR-ΔF508 (Fig. 3c and Supplementary Fig. 6a); SYN1-GFP (Supplementary Fig. 6b); and GFP-250 (Supplementary Fig. 6c). It is well known that CFTR-ΔF508 and GFP-250 form polyubiquitin-enriched aggresomes upon MG132 treatment and polyubiquitin-deficient aggresomes in an MG132-independent manner, respectively[22,39,41].

The immunostaining results revealed that when each of the CED components or HDAC6 was downregulated, MG132-induced aggresomes containing polypeptidyl-puro, CFTR-ΔF508 or SYN1-GFP were significantly dispersed (Fig. 3a and Supplementary Fig. 6a,b). Accordingly, the population of cells containing dispersed small cytoplasmic aggregates was increased by approximately two- to fourfold (Fig. 3b,c). In contrast, downregulation of the nonsense-mediated mRNA decay-specific factor, upstream frameshift 2 (UPF2), did not significantly affect aggresome formation (Fig. 3c and Supplementary Fig. 6a). Importantly, the polyubiquitin-deficient aggresome containing GFP-250 was not dispersed by CTIF downregulation (Supplementary Fig. 6c), indicating the preferential involvement of the CED complex in aggresome formation of polyubiquitinated polypeptides. Specific downregulation of the tested proteins by siRNAs was confirmed by western blotting (Supplementary Fig. 6d). Notably, the level of endogenous K63-ubiquitinated polypeptides was not significantly changed by CTIF downregulation (Supplementary Fig. 6e). These results indicate that the CED complex plays a critical role in efficient aggresome formation of polyubiquitin-enriched misfolded polypeptides.

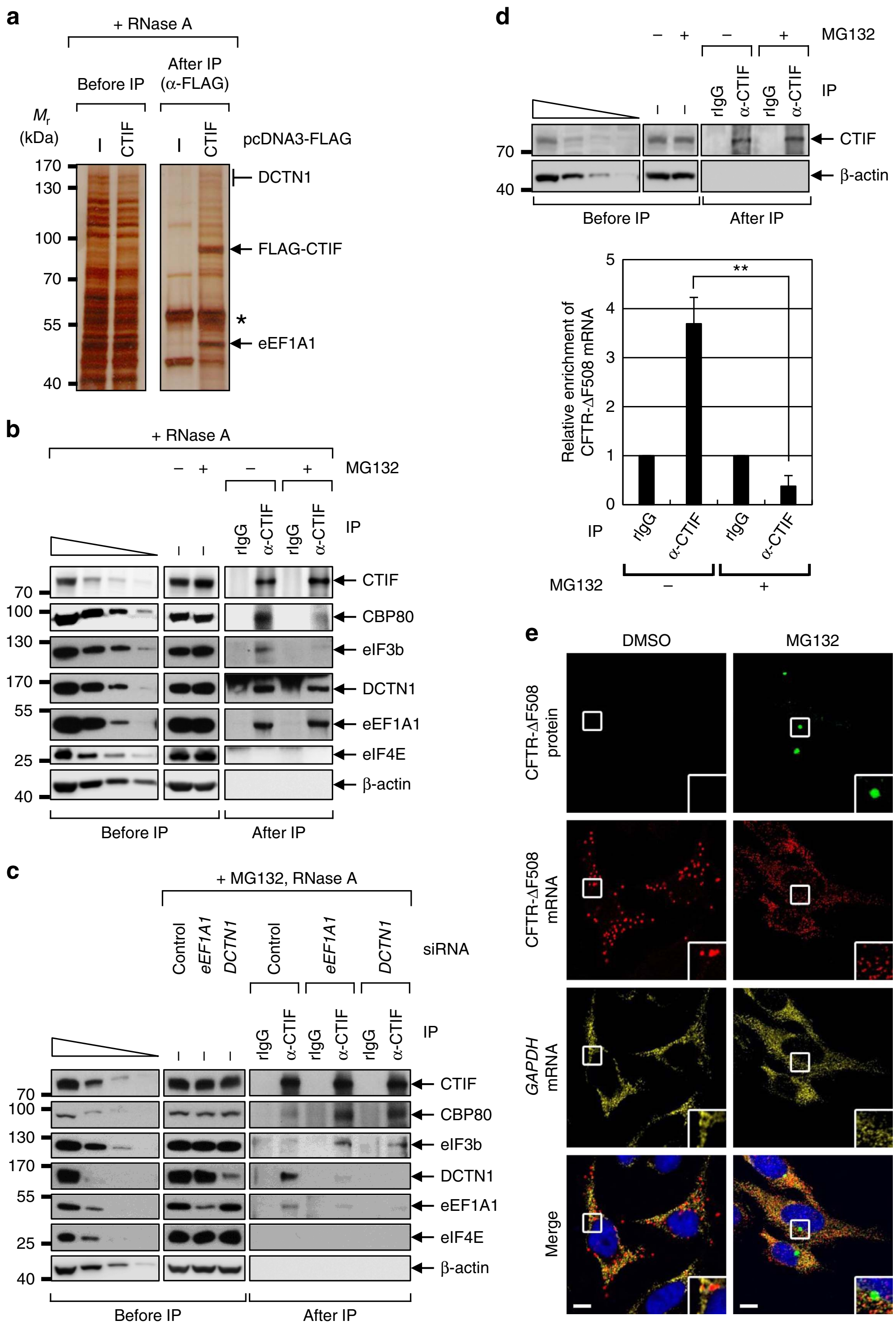


**Figure 2 | CTIF forms a complex with DCTN1 and eEF1A1 and is localized to the aggresome upon UPS impairment.** (**a**) Silver staining of immunoprecipitates of FLAG-CTIF in HEK293T cells. The indicated bands were excised and subjected to LC-MS/MS. Identified proteins in each band are specified. Immunoglobulin heavy chain is marked with an asterisk (*). MW, molecular weight. (**b**) IPs of endogenous CTIF. Extracts of HEK293T cells untreated or treated with MG132 for 12 h were obtained and then treated with RNase A before IP. IPs were performed using either α-CTIF antibody or nonspecific rabbit IgG (rIgG). Western blotting of samples before and after IPs was performed using the indicated antibodies. Threefold serial dilutions of total-cell extracts were loaded in the four left-most lanes. $n=2$. (**c**) IPs of endogenous CTIF using the extracts of HEK293T cells depleted of eEF1A1 or DCTN1. The cells were treated with MG132 for 12 h before harvesting. Total-cell extracts were treated with RNase A. $n=2$. (**d**) RNA IPs of endogenous CTIF. As in **b**, except that the extracts of HeLa cells stably expressing misfolded CFTR-ΔF508 were not treated with RNase A. Western blotting (upper) of CTIF and qRT–PCR of CFTR-ΔF508 mRNAs (lower) were performed using samples from either before or after IP. The levels of CFTR-ΔF508 mRNAs were normalized to the levels of *GAPDH* mRNA. The relative ratio of normalized CFTR-ΔF508 mRNAs obtained from the IPs using rIgG was arbitrarily set to 1.0. Columns and error bars represent the mean and s.d. of three independent transfections and qRT–PCRs. $n=3$. $^{**}P<0.01$. (**e**) Immunostaining for CFTR-ΔF508 protein (green), CFTR-ΔF508 mRNA (red) and *GAPDH* mRNA (yellow). $n=2$. Scale bar, 10 μm.

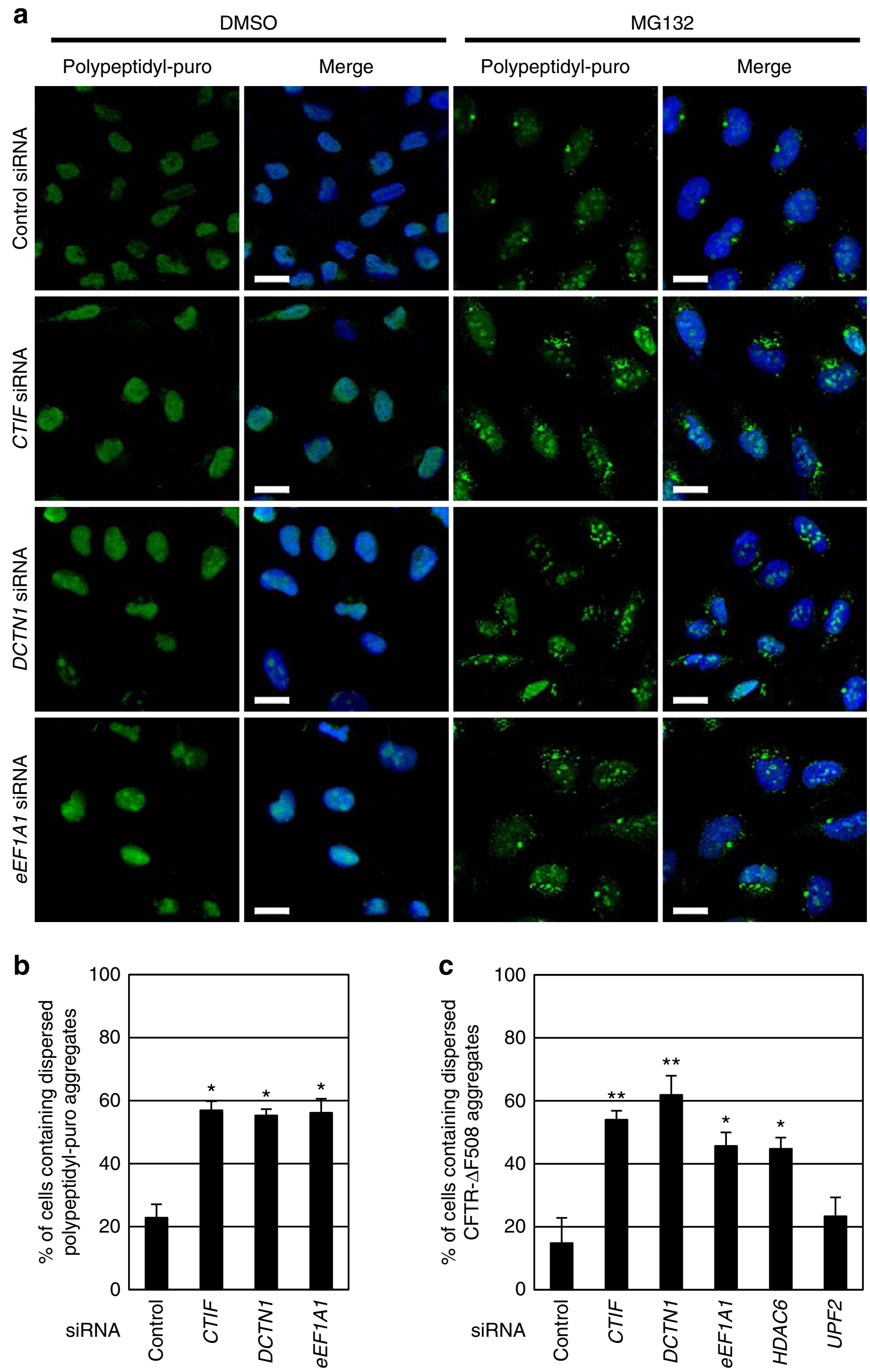


**Figure 3 | Aggresomes containing misfolded polypeptides are dispersed into small cytoplasmic aggregates by downregulation of CED components.** (**a**) Immunostaining of polypeptidyl-puro using α-puromycin antibody. HeLa cells were transfected with the indicated siRNA. Two days later, the cells were treated with either DMSO (left) or MG132 (right) for 12 h, and with puromycin for 1 h before cell staining. Scale bar, 20 μm. (**b**) Effect of CED downregulation on the formation of aggresomes containing misfolded polypeptidyl-puro. The cells containing aggresomes and dispersed aggregates of misfolded polypeptidyl-puro in **a** were counted. The relative percentage of cells containing dispersed aggregates was calculated. (**c**) Effect of CED downregulation on the formation of aggresomes containing misfolded CFTR-ΔF508. As performed in **a**,**b**, except that the cells containing aggresomes, and dispersed aggregates of CFTR-ΔF508 were counted. Immunostained images are shown in Supplementary Fig. 6a. $n = 2$. $^{*}P < 0.05$; $^{**}P < 0.01$.

**Association between CED and small cytoplasmic aggregates.** Several previous reports have shown that eEF1A1 can bind to both pre-existing misfolded polypeptides and newly synthesized polypeptides, and it generates a signal for aggresome formation[36,44,45]. We also observed a specific interaction between either eEF1A1 or CTIF and polypeptidyl-puro (Supplementary Fig. 7a–c), which typify newly synthesized misfolded polypeptides and DRiPs, because puromycin triggers premature termination of translation. Intriguingly, the observed interactions with polypeptidyl-puro were not affected by treatment with nocodazole, which inhibited aggresome formation and as a result triggered the accumulation of small cytoplasmic aggregates (Fig. 1d and Supplementary Fig. 2b). Furthermore, downregulation of eEF1A1 significantly inhibited the interaction of CTIF and polypeptidyl-puro (Supplementary Fig. 7d,e). These results indicate that (i) CTIF indirectly interacts with polypeptidyl-puro via eEF1A1 and (ii) the CED complex associates with misfolded polypeptides largely before aggresomal targeting and formation.

To further confirm the conclusion detailed above, we immunostained polypeptidyl-puro using the cells depleted of either CTIF or eEF1A1 in the presence of MG132 (Supplementary Fig. 8). The results showed that each CED component was co-localized with an aggresome containing polypeptidyl-puro upon MG132 treatment. Consistent with our observations that aggresomes are dispersed by the downregulation of CED components (Fig. 3 and Supplementary Fig. 6), the aggresome of polypeptidyl-puro was dispersed into the small cytoplasmic aggregates by CTIF downregulation. Intriguingly, the small cytoplasmic aggregates overlapped with each CED component. In a reciprocal experiment, the aggresome was also dispersed by eEF1A1 downregulation. However, CTIF partially or marginally overlapped with the small cytoplasmic aggregates of polypeptidyl-puro, supporting that CTIF associates with misfolded polypeptides via eEF1A1. These results suggest that CTIF is required for the efficient transport of misfolded polypeptides associated with eEF1A1.

**N-terminal region of CTIF is critical for CED formation**. To determine the minimal region within CTIF required for CED complex formation and aggresomal targeting, we performed a series of IPs using deletion variants of CTIF (Fig. 4a). The results revealed that both DCTN1 and eEF1A1 were significantly enriched in the IPs of FLAG-CTIF-WT, 12–598 and 1–305, but not of 54–598 and 101–598 (Fig. 4b). In agreement with a previous report that stem-loop–binding protein (SLBP) interacts with the C-terminal half of CTIF[46], SLBP was enriched in IPs of all tested proteins except FLAG-CTIF (1–305). Consistent with the IP results, FLAG-CTIF-WT and 12–598 were localized to the aggresome in the presence of either dimethyl sulfoxide (DMSO) or MG132 (Fig. 4c). In contrast, 54–598 and 101–598 exhibited perinuclear localization even in the presence of MG132, indicating that amino acids 12–53 of CTIF are critical for CED formation and aggresomal targeting of misfolded polypeptides.

To further demonstrate the minimal region of CTIF required for CED complex formation and aggresomal targeting of misfolded polypeptides, we constructed MYC-GST fusion proteins harbouring the N-terminal region of CTIF: MYC-CTIF(1–53)-GST and MYC-CTIF(12–53)-GST (Supplementary Fig. 9a). The GST pull-down results revealed that DCTN1 and eEF1A1 were enriched in the pull-down of MYC-CTIF (1–53)-GST and MYC-CTIF(12–53)-GST, but not MYC-GST (Supplementary Fig. 9b). Consistent with the GST pull-down results, MYC-CTIF(1–53)-GST and MYC-CTIF(12–53)-GST, but not MYC-GST, overlapped with γ-tubulin (Supplementary Fig. 9c). These results strongly indicate that a region spanning amino acids 12–53 of CTIF is critical and sufficient for CED formation and aggresomal targeting of misfolded polypeptides.

**Sequestration of CTIF into the aggresome inhibits CT**. Throughout this paper, we observed a tendency that, whereas CTIF was localized to the perinuclear region as well as the aggresome under normal conditions, the majority of perinuclear CTIF was redistributed to the aggresome upon MG132 treatment. In particular, deletion variants of CTIF lacking amino acids 1–53 exhibited strong localization to the perinuclear region and were not localized to the aggresome, compared with CTIF-WT (Fig. 4). These observations suggest the intriguing possibility that misfolded polypeptides may sequester the perinuclear CTIF into the aggresome and consequently reduce CT efficiency.

To examine the possibility mentioned above, we performed polysome fractionation assays and then analysed the relative distributions of CBP80 and eIF4E. It is well known that MG132 treatment causes the accumulation of misfolded polypeptides in the ER and activates PKR-like ER-localized eIF2α kinase, leading to an inhibition of general translation through eIF2α phosphorylation[47]. To minimize the inhibitory effect of MG132 on general translation via eIF2α phosphorylation, we employed mouse embryo fibroblast (MEF) cells expressing eIF2α mutant (A/A) harbouring the S51A substitution.

The results of polysome fractionation followed by western blotting showed that MG132 treatment disrupted polysome formation in a time-dependent manner (Supplementary Fig. 10a). Intriguingly, whereas the relative distribution of eIF4E was not detectably affected by MG132 treatment, a significant level of CBP80 was shifted from the polysome fractions to the subpolysome fractions upon MG132 treatment (Supplementary Fig. 10). Notably, the relative distributions of exogenously expressed GST-CTIF(54–598), which failed to form the CED complex (Fig. 4), remained unaffected upon MG132 treatment, compared with those of GST-CTIF-WT (Fig. 5). All the data indicate that misfolded polypeptides trigger the release of CTIF from the CT complex and, as a consequence, preferentially inhibit CT.

**CED renders cells more resistant to proteotoxic stresses**. The accumulation of misfolded and aggregated polypeptides is toxic to cells, as observed in various neurodegenerative diseases[22–24]. To determine whether the CED complex is involved in the cellular response to proteotoxic stresses induced by the accumulation of misfolded polypeptides, we investigated the effect of downregulation of CED components on apoptosis induced by overexpressed, misfolded CFTR-ΔF508 (Fig. 6a and Supplementary Fig. 11). Consistent with the results of a previous study[27], MG132 treatment caused minimal apoptosis (~2% of total cells) induced by stably expressed, misfolded CFTR-ΔF508. However, downregulation of one of the CED components or HDAC6 markedly promoted the CFTR-ΔF508-induced apoptosis of cells (~8–23% of total cells). In contrast, downregulation of UPF2 did not significantly affect misfolded CFTR-ΔF508-induced apoptosis. These results suggest that the CED complex renders cells more resistant to proteotoxic stresses induced by the accumulation of misfolded and aggregated polypeptides.

To more clearly demonstrate the role of the CED complex in apoptosis induced by the accumulation of misfolded CFTR-ΔF508, we carried out complementation experiments using HeLa cells stably expressing CFTR-ΔF508, *CTIF* siRNA and siRNA-resistant (R) FLAG-CTIF (FLAG-CTIF$^R$), either WT or 54–598, which fails to interact with DCTN1 and eEF1A1 (Fig. 4). Specific downregulation of CTIF by siRNA and comparable expression of FLAG-CTIF$^R$-WT and 54–598 were demonstrated by western blotting (Fig. 6b). As observed in Fig. 6a, downregulation of CTIF markedly increased the CFTR-ΔF508-induced apoptosis of cells (~27% of total cells). The observed increase in apoptosis was significantly reversed by the transient expression of FLAG-CTIF$^R$-WT but not of FLAG-CTIF$^R$(54–598; Fig. 6c,d), indicating that the interactions of CTIF with DCTN1 and eEF1A1 help cells to cope with proteotoxic stresses induced by the accumulation of misfolded polypeptide aggregates.

**CTIF is enriched in intracellular inclusion bodies**. Many neurodegenerative diseases are characterized by the presence of misfolded polypeptide-containing intracellular inclusion bodies[48–53]. For instance, the intracellular Lewy bodies containing aggregated α-synuclein, the inclusions of aggregated huntingtin protein and the inclusions of copper–zinc superoxide dismutase (SOD1) mutants are histological hallmarks found in Parkinson's disease (PD), Huntington's disease and amyotrophic lateral

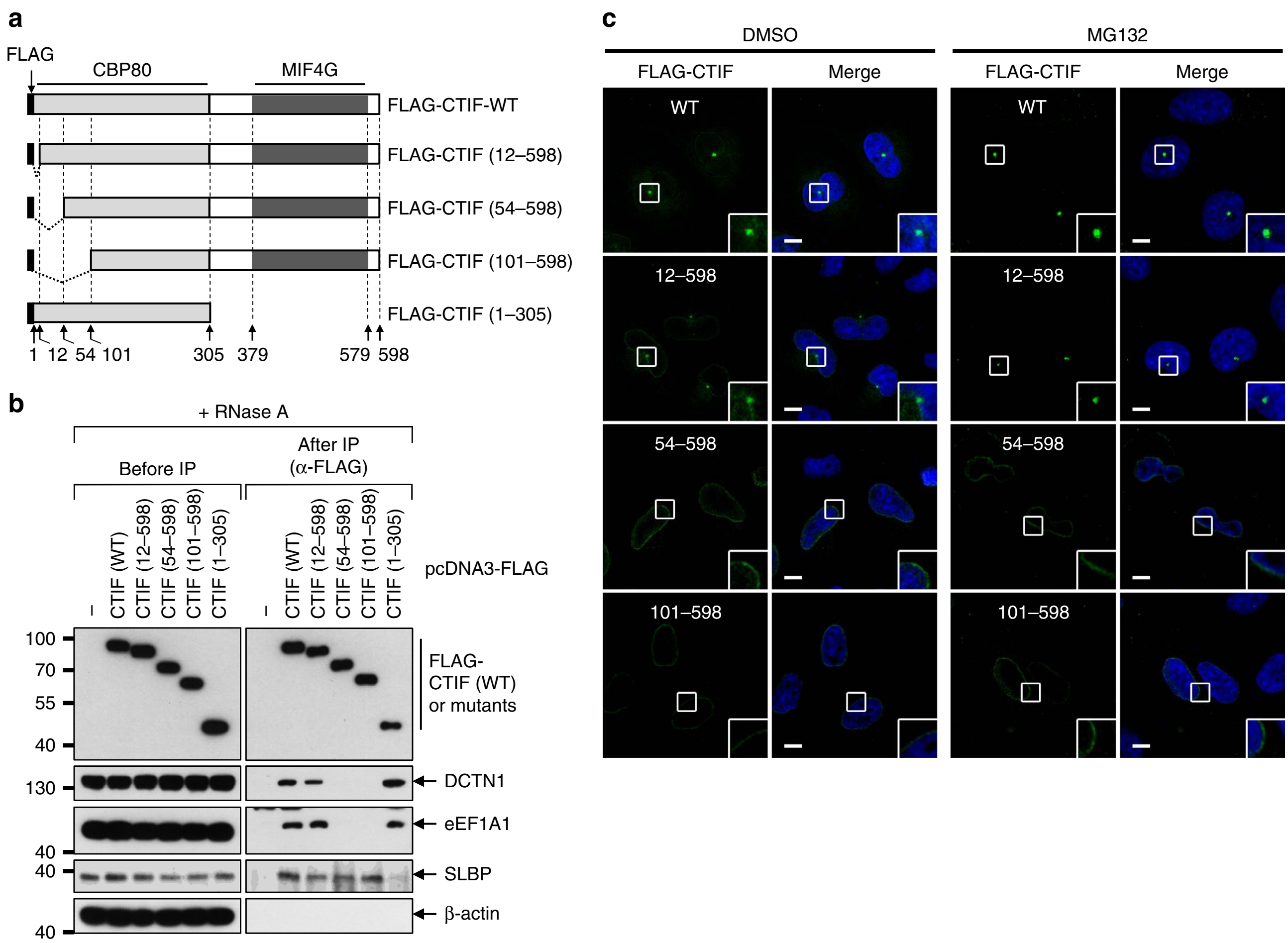


**Figure 4 | N-terminal region spanning amino-acid residues 12-53 of CTIF is critical for CED complex integrity and aggresomal targeting.** (**a**) Schematic diagram of FLAG-tagged CTIF variants. CBP80-interacting region and MIF4G domain are specified by light grey and dark grey boxes, respectively. (**b**) CTIF domain mapping by IPs using N-terminal serial deletions of CTIF in HEK293T cells. (**c**) Immunostaining of FLAG-tagged CTIF variants (green) in HeLa cells in the presence of either DMSO or MG132. $n = 3$. Scale bar, 10 μm.

sclerosis, respectively. The intracellular inclusion bodies are biochemically and morphologically similar to the aggresome[48], and the formation of inclusion bodies is driven by a cellular process that is analogous to the formation of aggresomes[27]. Therefore, our results, which indicated the important role of CED complex in aggresome formation, led us to examine whether CTIF is enriched in the intracellular inclusion bodies of neurodegenerative diseases. To this end, we first determined whether CTIF is co-localized with inclusion bodies containing an amyotrophic lateral sclerosis-linked G93A mutant of SOD1 (SOD1-G93A)[54,55]. Immunostaining showed that SOD1-G93A, but not SOD1-WT, overlapped with CTIF in a MG132-dependent manner (Supplementary Fig. 12).

We also performed immunohistochemical analysis using brain sections from PD patients with an age-matched normal subject as a control and antibodies against CTIF and α-synuclein (a marker protein of Lewy bodies; Fig. 7a). Compared with the normal control brain, Lewy bodies were identifiable with larger α-synuclein-immunoreactive bodies throughout the cerebellar molecular layer of the PD patient brain sections (Fig. 7a, α-synuclein panel). Strong CTIF immunoreactivity was also observed within large inclusion bodies in the cerebellar molecular layer (ml) in the brains of PD patients (Fig. 7a, CTIF panel). Using double immunofluorescence and confocal microscopic analysis, we found that CTIF-immunoreactive structures in PD patients co-localized with α-synuclein-immunoreactive bodies, which were not detected in sections from the normal brain (Fig. 7b). These data indicate that CTIF might be a component of Lewy bodies.

## Discussion

The presence of intracellular inclusion bodies containing misfolded polypeptides is a hallmark of many neurodegenerative diseases. Despite the importance of understanding the strategies used by cells to cope with misfolded polypeptides, little is known regarding how misfolded polypeptides form aggresomes and are eventually eliminated from cells by aggrephagy. Here we found that the CED complex plays a pivotal role in monitoring the quality of misfolded polypeptides. On the basis of these results, we propose a model for how eukaryotic cells cope with misfolded polypeptides (Fig. 7c).

Newly synthesized mRNA with a 5′-cap bound to CBC is exported from the nucleus to the cytoplasm. During or after export, the mRNA recruits ribosomes through CBC with the help of translation initiation factors, including CTIF, eIF4AIII and eIF3 (refs 2,4,6,8,9,56). After CBC replacement with eIF4E, polypeptides are synthesized in the cytoplasm in bulk amounts. If the newly synthesized polypeptides resulting from either CT or ET are properly folded, and therefore considered normal, they are correctly targeted to their precise intracellular locations and perform their intended functions. If misfolded, however,

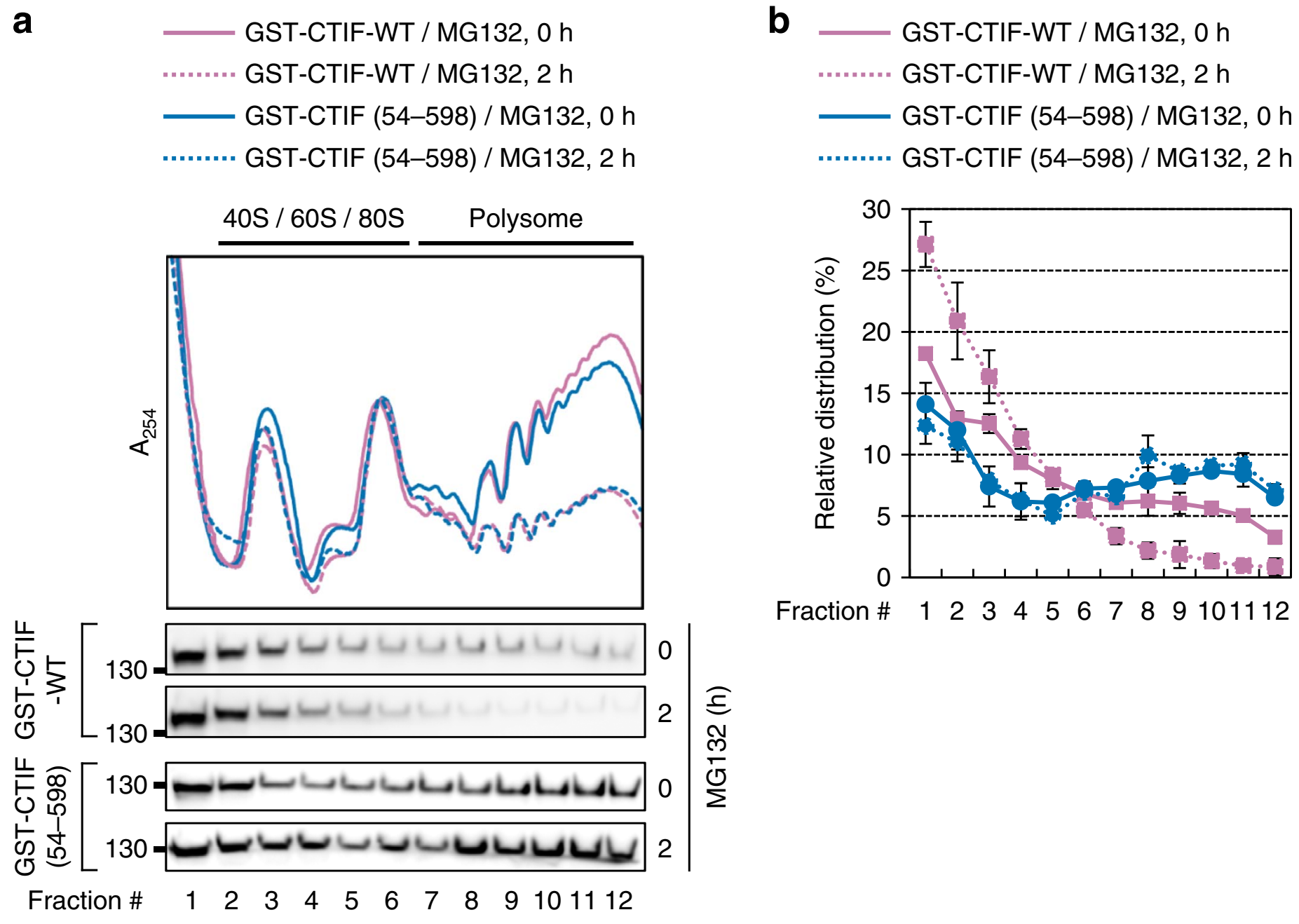


**Figure 5 | Accumulation of misfolded polypeptides preferentially inhibits CT.** (**a**) Polysome fractionation analysis of MEF-eIF2α (A/A) cells. Cells transiently expressing GST-CTIF-WT or GST-CTIF(54-598) were treated with MG132 for the indicated time before harvesting. Each fraction was subjected to western blotting using the indicated antibodies. (**b**) Relative distributions of GST-CTIF-WT or GST-CTIF(54-598) in polysome fractions. The intensities of each band in **a** were quantitated, and then the relative levels of GST-CTIF-WT or GST-CTIF(54-598) in each fraction were calculated as a percentage of the total. The dots and error bars represent the mean and s.d. of two biological replicates.

the polypeptides are rapidly degraded by the UPS[12–15]. However, when the UPS is impaired or overwhelmed, misfolded polypeptides associate with the CED complex via a direct association with eEF1A1 (Supplementary Figs 7 and 8), resulting in two molecular events. First, the misfolded polypeptides are released in association with the CED complex from the CT complex (Fig. 2), are transported to an aggresome via a DCTN1-mediated retrograde movement (Figs 3 and 4) and are eventually degraded by aggrephagy. Second, aggresomal targeting of misfolded polypeptides is accompanied by the release of CTIF from the CT complex, resulting in preferential inhibition of CT (Fig. 5), which might contribute to a reduction in the level of misfolded polypeptides within the cells.

How does the misfolded polypeptide-associated CED complex activate dynein-mediated retrograde transport? The association between the CED complex and misfolded polypeptides would trigger a marked remodelling of the CED complex, leading to a conformational change in DCTN1 from an unfavourable to a favourable structure for activating the dynein motor. The favourable structure of DCTN1 may drive efficient recruitment of dynein and promote processivity of the dynein motor. Indeed, recent structural and single-molecule experiments have revealed that, although DCTN1 contains a dynein-binding motif, a stable interaction between the mammalian dynein motor and dynactin requires a cargo adaptor, which links the motor complex to the cargo[57,58]. Formation of the ternary complex (dynactin–dynein–cargo adaptor) activates the dynein motor and converts it from a nonprocessive to a highly processive motor[57,58].

Several translation-coupled protein quality-control mechanisms have been characterized, which are typified by the ribosome quality control complex (RQC)[16,59]. The RQC selectively removes abnormal nascent polypeptides being synthesized from stalled ribosomes, rather than pre-existing or newly synthesized misfolded polypeptides. Although it is unclear which step of translation is linked to translation-coupled protein quality-control mechanisms including RQC, it may be advantageous for cells to sense the synthesis of misfolded polypeptides as early as possible. In this respect, the CTIF-mediated coupling of CT and CED-mediated protein surveillance would be more beneficial for cell proliferation and survival. In agreement with this, downregulation of CTIF promoted apoptosis induced by an accumulation of misfolded CFTR-ΔF508 (Fig. 6).

RQC and CED play a common role in the degradation of abnormal polypeptides: nascent misfolded polypeptides and either pre-existing or newly synthesized misfolded polypeptides, respectively. However, it is likely that the pathways involving these two complexes differ mechanistically. Indeed, our data showed that CED-mediated aggresome formation was not affected by the downregulation of listerin 1 (LTN1), an E3 ligase (Supplementary Fig. 13), which plays a critical role in RQC-mediated polypeptide degradation by ubiquitinating nascent polypeptides[60]. Therefore, CED-mediated protein surveillance is a previously unappreciated protein quality-control mechanism in mammalian cells.

HDAC6 has been identified as an adaptor molecule that links polyubiquitinated small cytoplasmic aggregates and dynein motor proteins[27,30,31]. When the UPS is impaired or overwhelmed, polyubiquitinated small cytoplasmic aggregates associate with HDAC6 via their unanchored ubiquitins or ubiquitin chains[32], and are then transported to aggresomes in a microtubule- and dynein-dependent manner. The CED complex characterized in this study might cooperate with HDAC6 to promote efficient aggresome formation and the aggrephagy of misfolded polypeptides upon UPS impairment. Multiple lines of evidence in this study support the cooperativity of the CED complex and HDAC6. First, CTIF was co-localized with HDAC6 (Supplementary Fig. 3a). Second, whereas a small amount of HDAC6 was detected in the IP of CTIF, MG132 treatment promoted the association between HDAC6 and CTIF (Supplementary Fig. 5a). Third, downregulation of either CED

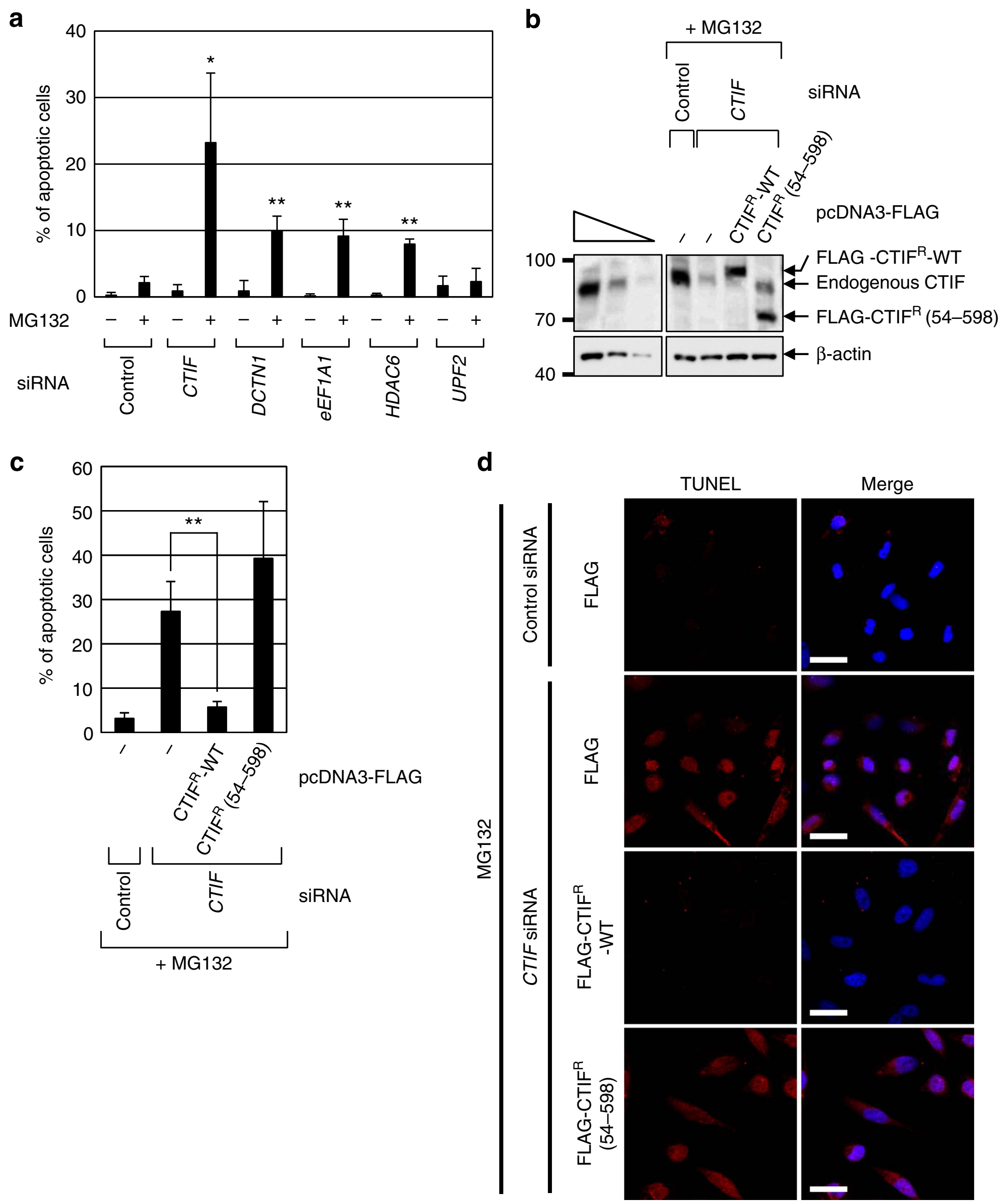


**Figure 6 | Downregulation of CED components promotes apoptosis of the cells induced by accumulation of stably expressed misfolded CFTR-ΔF508.** (**a**) TUNEL assay of HeLa cells stably expressing misfolded protein CFTR-ΔF508. HeLa cells were transfected with the indicated siRNA. Two days later, the cells were untreated or treated with MG132 for 12 h, and then analysed by a TUNEL assay (Supplementary Fig. 11). The ratio of stained cells was calculated and is represented as a percentage. (**b–d**) Complementation assay using siRNA-resistant (R) CTIF variants. HeLa cells stably expressing misfolded protein CFTR-ΔF508 were transfected with either *CTIF* siRNA or nonspecific control siRNA. One day later, cells were re-transfected with a plasmid expressing FLAG-CTIF$^R$-WT or FLAG-CTIF$^R$(54-598). One day later, the cells were treated with MG132 for 12 h, and then analysed by western blotting (**b**) and a TUNEL assay (**d**). Scale bar, 50 µm. The ratio of stained cells was calculated and is represented as a percentage (**c**). $n = 3$, $^*P < 0.05$; $^{**}P < 0.01$.

components or HDAC6 commonly triggered the dispersion of aggresomes containing polyubiquitin-enriched CFTR-ΔF508, but not polyubiquitin-deficient GFP-250, into small cytoplasmic aggregates (Fig. 3c and Supplementary Fig. 6a,c). These observations suggest that the CED complex functions cooperatively with HDAC6 to trigger the aggresome formation of misfolded and polyubiquitinated proteins.

In addition to CED and HDAC6, the BCL2-associated athanogene 3 (BAG3)-containing chaperone/co-chaperon complex is known to recognize misfolded polypeptides and transport them to aggresomes[61,62]. We observed that (i) MG132 treatment upregulated BAG3, consistent with a previous report[62], and promoted the association between endogenous BAG3 and FLAG-CTIF (Supplementary Fig. 14a), and (ii) the polyubiquitin-enriched aggresome containing misfolded CFTR-ΔF508 was significantly dispersed by BAG3 downregulation (Supplementary Fig. 14b,c). Therefore, it is most likely that CED may physically interact with and act in concert with HDAC6 and BAG3 for efficient aggresome formation.

Clinically, aggresomes have received much attention owing to their biochemical and morphological similarities to the misfolded protein inclusion bodies observed in the cytoplasm of neuronal cells affected by many neurodegenerative diseases. We also observed that CTIF is considerably enriched in the Lewy bodies found in neurons affected by PD (Fig. 7a,b). In light of their similarities, therefore, aggresomes are thought to play a critical role in both protein surveillance and the pathogenesis of neurodegenerative diseases. Our findings for CED-mediated

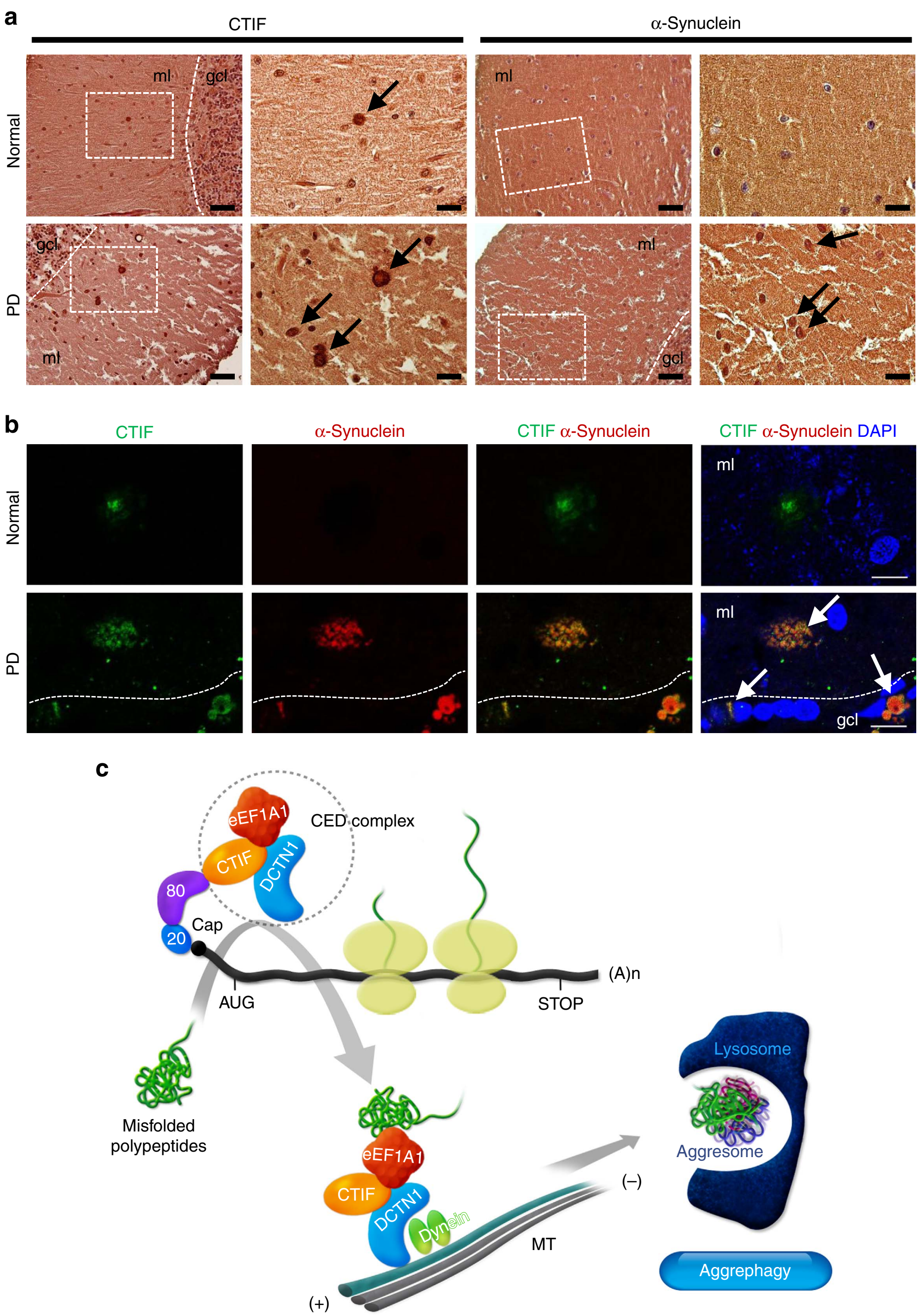


**Figure 7 | CTIF is enriched in the Lewy bodies of PD.** (**a**) Immunohistochemistry for CTIF and α-synuclein in the cerebellar molecular layer of PD patients. A normal subject was examined as a control. Nuclei were visualized with haematoxylin and eosin (H&E) counterstain (haematoxylin-positive nuclei, purple). Right panels are higher-magnification views of the left panels (dashed boxes). Black arrows indicate Lewy body-like structure. Scale bar, 50 μm (left), 20 μm (right). ml, molecular layer; gcl, granular cell layer. (**b**) Double-immunofluorescent histochemistry for CTIF (green) and α-synuclein (red) in the cerebellar molecular layer of PD patients. Nuclei (blue) were visualized using DAPI. White arrows indicate double-labelled cells. Scale bar, 10 μm. (**c**) Proposed model for CED-mediated protein surveillance and CT inhibition. The details are described in the Discussion section.

protein surveillance and aggresome formation could provide molecular insight into the treatment of neurodegenerative disease.

## Methods

**Plasmid construction.** The following plasmids have been described previously: pcDNA3-FLAG-CTIF-WT, pcDNA3-FLAG-CTIF(1–305) and pCMV-Myc-CTIF[6]; pcDNA3-FLAG-Dcp1a[63]; and pcDNA3-FLAG[6,64].

The following plasmids were kindly provided as follows: pcDNA3-HDAC6-FLAG provided by Tso-Pang Yao[27]; pCXbsr-mRFP-Ub and pEGFP-N1-SYN1 by Michael Sherman[36,39]; GFP-CFTR-ΔF508 (CFTR-ΔF508) by Ryan Tyler[27]; GFP-250 by Elizabeth Sztul[65]; and pEBG expressing GST by Sung Key Jang (Pohang University of Science and Technology, Pohang, Korea).

The plasmids pCNS-D2-dynactin 1 containing DCTN1 cDNA (NM_004082.4) and pOTB7-eEF1A1 containing full-length human eEF1A1 cDNA (NM_001402.5) were purchased from the Korea Human Gene Bank (Daejeon, Korea).

To construct pCMV-Myc-DCTN1, pCMV-Myc (Clontech) was digested with EcoRI and Acc65I and then ligated to two PCR-amplified fragments: an EcoRI/BamHI fragment of the PCR-amplified 5′-half of DCTN1 cDNA and a BamHI/Acc65I fragment of the PCR-amplified 3′-half of DCTN1 cDNA. The EcoRI/BamHI fragment was amplified using pCNS-D2-dynactin 1 and two oligonucleotides: 5′-AAATATGCGGCCGCCCGAATTCGCATGGCACAGAGCA AGAGGCACGTGTAC-3′ (sense) and 5′-GATACCTGTGGTCCAAAGGCCA GTGCAGC-3′ (antisense). A BamHI/Acc65I fragment was amplified using pCNS-D2-dynactin 1 and two oligonucleotides: 5′-GCAAGAAGATCCGAAGGC GAATGCCAGG-3′ (sense) and 5′-GGGGTACCTTAGGAGATGAGGCGACTGT GAAGCTGG-3′ (antisense), where the underlined nucleotides specify the EcoRI and BamHI sites. PCR amplification was carried out using the Advantage-HF2 PCR Kit (Clontech).

To construct plasmids p3 × FLAG-eEF1A1 and pCMV-Myc-eEF1A1, which encode FLAG-tagged and Myc-tagged full-length human eEF1A1 cDNA, respectively, a NotI/KpnI fragment from p3 × FLAG-CMV-7.1 (Sigma) and an EcoRI/KpnI fragment from pCMV-Myc (Clontech) were ligated to a PCR-amplified fragment that contained full-length human eEF1A1 cDNA and was digested with NotI/KpnI and EcoRI/KpnI, respectively. The full-length human eEF1A1 cDNA was amplified using pOTB7-eEF1A1 as a template and two oligonucleotides 5′-CGGAATTCCGGCGGCCGCTATGGGAAAGGAAAAGACT CATATCAACATTG-3′ (sense) and 5′-GGGGTACCTCATTTAGCCTTCTGAGC TTTCTGGGCAG-3′ (antisense), where the underlined nucleotides specify the EcoRI, NotI and KpnI sites, respectively.

For bacterial production of human DCTN1 and eEF1A1, the EcoRI/Klenow-filled Acc65I fragment from pCMV-Myc-DCTN1 and the KpnI/Klenow-filled NotI fragment from p3 × FLAG-eEF1A1 were inserted into pRSET A (Invitrogen), respectively.

To construct plasmids expressing a series of N-terminal deletions of CTIF, a BamHI/BstEII fragment of pcDNA3-FLAG-CTIF-WT containing the 5′-terminal region of the full-length CTIF cDNA was replaced with a PCR-amplified fragment that contained a 5′-terminal deletion of CTIF cDNA and was digested with BamHI and BstEII. The 5′-terminal deletion fragment of CTIF cDNA was amplified using pcDNA3-FLAG-CTIF-WT as a template and two oligonucleotides: 5′-CGGGATCC GAGGCAGGGAGCAGCCGCTCCCAGGAG-3′ (sense) and 5′-GTGTTTGGCAC TGCCTGGCTGGTGGTC-3′ (antisense) were used to construct pcDNA3-FLAG-CTIF(12–598); 5′-CGGGGATCCACCCAGTCCCACATCTCCCAGTGGAC-3′ (sense) and 5′-GGCACTGCCTGGCTGGTGGTCACC-3′ (antisense) were used to construct pcDNA3-FLAG-CTIF(54–598); and 5′-CGGGGATCCGCGGCC AACACCTTCGATTCCTTC-3′ (sense) and 5′-GGCACTGCCTGGCTGGT GGTCACC-3′ (antisense) were used to construct pcDNA3-FLAG-CTIF(101–598), respectively. The underlined nucleotides specify the BamHI and BstEII sites, respectively.

To construct pCMV-Myc-CTIF(1–53)-GST and pCMV-Myc-CTIF(12–53)-GST, a BglII/NotI fragment of pCMV-Myc was ligated to two fragments: (i) a PCR-amplified and BglII/XhoI-digested fragment containing a CTIF(1–53) or CTIF(12–53) sequence; and (ii) a XhoI/NotI fragment of pEBG containing the GST sequence. The PCRs were performed using pcDNA3-FLAG-CTIF-WT as a template and two oligonucleotides: 5′-GGAAGATCTCTATGGAAAACTCCTCT GCAGCATCAGCC-3′ (sense) and 5′-CCGCTCGAGCCTCTCGCTCTCGCCATC ACCCTCC-3′ (antisense) were used to construct pCMV-Myc-CTIF(1–53)-GST; and 5′-GGAAGATCTCTGAGGCAGGGAGCAGCCGCTCCCAG-3′ (sense) and 5′-CCGCTCGAGCCTCTCGCTCTCGCCATCACCCTCC-3′ (antisense) were used to construct pCMV-Myc-CTIF(12–53)-GST. The underlined nucleotides specify the BglII and XhoI sites, respectively. To generate pCMV-Myc-GST, a Klenow-treated SalI/XhoI fragment of pCMV-Myc-CTIF(1–53)-GST was self-ligated.

To construct pEBG-CTIF-WT and pEBG-CTIF(54–598), the BamHI/Klenow-filled ClaI fragment from pEBG was ligated to the BamHI/EcoRV fragment from pcDNA3-FLAG-CTIF and pcDNA3-FLAG-CTIF(54–598), respectively.

To construct pcDNA3-FLAG-CTIF$^R$-WT, which encodes an siRNA-resistant (R) version of FLAG-CTIF, specific target sequences that anneal to *CTIF* siRNA were mutated from 5′-GAAGTGGAGATCGCACACA-3′ to 5′-GA*G*GT*A*GA*A*A T*A*GC*G*CA*C*A-3′, where italicized nucleotides specify the sites of silent mutations that confer resistance to *CTIF* siRNA. To construct pcDNA3-FLAG-CTIF$^R$ (54–598), a BamHI/BstEII fragment of pcDNA3-FLAG-CTIF$^R$-WT was replaced with a BamHI/BstEII fragment, which contained an N-terminal deletion fragment of CTIF cDNA and was amplified by PCR. The PCR was performed using pcDNA3-FLAG-CTIF$^R$-WT as a template and two oligonucleotides: 5′-CGG GGATCCACCCAGTCCCACATCTCCCAGTGGAC-3′ (sense) and 5′-GGCACT GCCTGGCTGGTGGTCACC-3′ (antisense). The underlined nucleotides specify the BamHI and BstEII sites, respectively.

**Cell culture and transfection and cell line generation.** HeLa and HEK293T cells (purchased from ATCC) were cultured in DMEM (Hyclone) containing 10% fetal bovine serum (Hyclone) and 1% penicillin/streptomycin (Hyclone). MEF cells expressing eIF2α mutant (A/A) harbouring S51A substitution (a kind gift from Sung Hoon Back, Ulsan University, Republic of Korea) were maintained in DMEM supplemented with 10% fetal bovine serum, 1% penicillin/streptomycin and 1 × MEM non-essential amino acids (Gibco). Cells were transiently transfected with the indicated plasmids using calcium phosphate or Lipofectamine 2000 (Invitrogen).

HeLa cells stably expressing CFTR-ΔF508 were generated by transfection of a plasmid expressing GFP-CFTR-ΔF508 with Lipofectamine 2,000. Two days after transfection, the cells were serially diluted and cultured in DMEM containing 0.8 mg ml$^{-1}$ Geneticin (G418; Gibco). Colonies were selected and maintained in DMEM containing 0.4 mg ml$^{-1}$ G418. Stable expression of CFTR-ΔF508 was confirmed by immunostaining and quantitative real-time PCR (qRT–PCR). All cell lines were regularly cultured in DMEM containing plasmocin (Invivogen) and tested for mycoplasma contamination using the MycoAlert PLUS Mycoplasma detection kit (Lonza).

**siRNA transfection.** To downregulate endogenous protein using a specific siRNA, the cells were transfected with 100 nM *in vitro*-synthesized siRNA (GenePharma) using Oligofectamine (Invitrogen). The siRNA sequences for the control[66], *UPF2* (ref. 66) and *CTIF*[6] have been described previously. Endogenous DCTN1, eEF1A1 and HDAC6 were downregulated using 5′-r(CAGAGAAGGCAGAACUAAA) d(TT)-3′, 5′-r(CCCAGGACACAGAGACUUU)d(TT)-3′ and 5′-r(AGACCUAA UCGUGGGACUGC)d(TT)-3′, respectively. To downregulate endogenous LTN1 and BAG3, two different siRNAs targeting the same gene were used: 5′-r(GGA AGAAAGAGAAGCUAAA)d(TT)-3′ for LTN1–1, 5′-r(AGCCAAACCUCUUG AAAUA)d(TT)-3′ for LTN1-2, 5′-r(AAGGUUCAGACCAUCUUGG)d(TT)-3′ for BAG3-1 and 5′-r(UUUCUUCUAUAUUCUUACU)d(TT)-3′ for BAG3-2.

**Chemical treatments.** Where indicated, the cells were treated with the following chemicals: MG132 (5 μM; Calbiochem), bafilomycin A1 (100 nM; Calbiochem), nocodazole (1 μM; Calbiochem) and DMSO (BioShop) for 12 h. To observe polypeptidyl-puro, the cells were treated with MG132 for 12 h and treated with puromycin (1 μg ml$^{-1}$ for immunostaining and 10 μg ml$^{-1}$ for IPs; Sigma) for 1 h before cell fixation or harvesting.

**RNA preparation and quantitative real-time PCR.** Total RNAs were purified using TRIzol Reagent (Invitrogen). qRT–PCR analyses were performed with cDNA and gene-specific oligonucleotides with a LightCycler 480 SYBR Green I Master Kit (Roche Diagnostics GmbH) using a LightCycler 480 II machine. The following gene-specific oligonucleotides were used: 5′-GGAGTACAACTACAACAGCC-3′ (sense) and 5′-CAGCAGGACCATGTGATCGC-3′ (antisense) to detect CFTR-ΔF508 mRNAs; and 5′-TGGCAAATTCCATGGCACC-3′ (sense) and 5′-AGA GATGATGACCCTTTTG-3′ (antisense) to detect *GAPDH* mRNAs.

**Fluorescence *in situ* hybridization.** To determine the intracellular distributions of mRNAs, HeLa cells stably expressing CFTR-ΔF508 were analysed using a QuantiGene ViewRNA *in situ* hybridization Cell Assay (Affymetrix) according to the manufacturer's instructions. Briefly, the cells were fixed with 4% paraformaldehyde (Sigma) in PBS for 30 min at room temperature (RT) and then permeabilized with detergent solution for 10 min at RT. Next, the cells were treated with protease for 10 min at RT, and then treated with specific probes capable of hybridizing to either GFP mRNA or *GAPDH* mRNA for 3 h at 40 °C. After mRNA–probe hybridization, the cells were incubated for 30 min at 40 °C with PreAmplifier Mix, Amplifier Mix and Label Probe Mix, sequentially. After the RNA-fluorescent *in situ* hybridization assay, the GFP-positive aggresomes were stained as described in the immunostaining section.

**Antibodies.** Antibodies against the following molecules were used for immunostaining, IPs and western blotting: CTIF[6] (1:50 for immunostaining and 1:1,000 for western blotting), CBP80 (ref #8 or #9983, Cell Signaling, 1:1,000), eIF3b (sc-16377, Santa Cruz, 1:1,000), DCTN1 (p150$^{glued}$; 610474, BD Biosciences, 1:12.5 for immunostaining and 1:250 for western blotting), eEF1A1 (CBP-KK1; #05–235, Merck Millipore, 1:1,000), eIF4E (C46H6; #2067, Cell Signaling, 1:1,000 or 610269, BD Biosciences, 1:500), β-actin (A5441, Sigma, 1:10,000), GST (A190-122A, Bethyl Laboratories, 1:8,000), His (27-4710-01, GE Healthcare, 1:3,000), FLAG (OctA-Probe (D-8), 1:40; sc-807, Santa Cruz (rabbit polyclonal) and F1804, Sigma (mouse monoclonal), 1:80 for immunostaining), FLAG M2-Peroxidase (horseradish peroxidase; A8592, Sigma,1:5,000 for western blotting), HDAC6 (sc-11420, Santa Cruz, 1:1,000), eIF4AIII[67] (1:1,000), rpS3 (ab140688, Abcam, 1:20,000), eIF4GI (a gift from S. K. Jang (Pohang University of Science and Technology, Pohang, Korea), 1:5,000), UPF2 (ref. 67; 1:1,000), γ-tubulin (sc-17788, Santa Cruz, 1:20), Myc (9E10; OP10L, Calbiochem, 1:1,000 for western blotting and sc-789, Santa Cruz, 1:40 for immunostaining), Vimentin (v5255, Sigma, 1:200), DCP1A (D5444, Sigma, 1:50), GFP (sc-9996, Santa Cruz, 1:50), puromycin (12D10; #MABE343, Merck Millipore, 1:400 for immunostaining and 1:10,000 for western blotting), LTN1 (OAAB11604, Aviva, 1:500), SLBP[68] (1:2,000), K63-linkage-specific polyubiquitin (#5621, Cell Signaling, 1:1,000), α-Synuclein (610787, BD Biosciences, 1:50), GAPDH (LF-PA0212, AbFrontier, 1:10,000), BAG3 (10599-1-AP, Proteintech, 1:5,000), Alexa Fluor 488 goat α-mouse IgG (A-11017, Invitrogen, 1:200), rhodamine-conjugated goat α-rabbit IgG (31670, Invitrogen, 1:200), biotinylated goat α-Rabbit IgG (BA-1000, Vector Laboratories, 1:500), biotinylated goat α-mouse IgG (BA-9200, Vector Laboratories, 1:500), Alexa Fluor 568 donkey α-mouse IgG (A10037, Invitrogen, 1:1,000) and Alexa Fluor 488 donkey α-rabbit IgG (H + L; A-21206, Invitrogen, 1:500).

**Immunostaining.** Immunostained HeLa cells were observed using either an LSM 510 Meta or an LSM 700 confocal microscope (Carl Zeiss). HeLa cells were fixed with 2% paraformaldehyde in PBS for 10 min and then permeabilized with 0.5% Triton X-100 in PBS for 10 min. The cells were incubated with 1.5% BSA (Santa Cruz) in PBS for 1 h and then with primary antibodies in PBS containing 0.5% BSA for 1 h. Next, the cells were incubated with Alexa Fluor 488 or rhodamine-conjugated secondary antibodies for 1 h, and nuclei were stained with 4′,6-diamidino-2-phenylindole (DAPI; Biotium) for 5 min. The cells were then mounted (Dako) and observed using a Zeiss confocal microscope. All procedures were performed at RT.

**Quantification of aggresome-containing cells.** To quantify CTIF bodies, 100 cells containing CTIF bodies were counted and the size of each CTIF body was measured using a Zeiss LSM Image Browser. To quantify the cells containing aggresomes of polypeptidyl-puro or CFTR-ΔF508, 50–100 cells were counted. Cell counting was performed in a blinded way by two experienced independent investigators. The results obtained from at least two biological replicates were independently counted.

**IP and RNA-IP.** IP and RNA-IP were performed using HEK293T cells or HeLa cells stably expressing CFTR-ΔF508. Where indicated, cells were transfected with the indicated plasmids by calcium phosphate precipitation. Two days after transfection, cells were washed using ice-cold PBS and harvested by centrifugation at 3,000 *g* for 10 min at 4 °C. The pellet was resuspended with 500 μl of NET-2 buffer (1 mM phenylmethylsulfonyl fluoride (PMSF), 2 mM benzamidine, 150 mM NaCl, 0.05% NP-40 and 50 mM Tris-HCl (pH 7.4)). And then, the cells were sonicated two times with 30 bursts of 1 s each (Branson Sonifier 250, output control 3, duty cycle 30%). Cell extracts were centrifugated at 13,800 *g* for 10 min at 4 °C. The supernatants were pre-cleared with 50 μl of protein G or A-agarose beads (Incosphram) for 90 min at 4 °C. While pre-clearing the supernatant, a specific antibody for IP was incubated with 50 μl of protein G or A-agarose beads for 90 min at 4 °C, and antibody-conjugated beads were washed with ice-cold NET-2 buffer. Pre-cleared supernatants were mixed with antibody-conjugated bead and incubated for 2 h at 4 °C. In the case of IP of FLAG-tagged protein, FLAG antibody-conjugated agarose beads (Sigma) were added. After incubation, the beads were washed five times and suspended in 100 μl of 2 × sample buffer (10% β-mercaptoethanol, 4% SDS, 100 mM Tris-HCl (pH 6.8), 15% glycerol and 0.008% Bromophenol Blue). Co-immunopurified proteins were analysed by western blotting.

RNA-IP was performed similarly to IP as described above, except that the supernatants obtained after sonication were incubated with tRNA-saturated and antibody-conjugated beads. After RNA-IP, the beads were suspended in 100 μl of 2 × sample buffer. After quick spin-down, 80 μl of the supernatants was used for the detection of co-immunopurified RNAs. The RNAs were extracted using phenol, chloroform and isoamyl alcohol, and precipitated using ethanol. The remaining supernatants were used for western blotting.

**Western blotting.** Cell extracts or immunopurified proteins were subjected to SDS–polyacrylamide gel electrophoresis. And then proteins were transferred to Protran Premium nitrocellulose (Amersham) and were probed by a specific antibody.

**Polysome fractionation.** MEF-eIF2α (A/A) cells were cultured in three 150-mm culture dishes. When indicated, the cells were transiently transfected with plasmid expressing either GST-CTIF-WT or GST-CTIF(54–598). Two days after transfection, cells were washed with 10 ml of ice-cold PBS containing 100 $\mu g\, ml^{-1}$ cycloheximide. After washing, cell extraction was resuspended with 1 ml of lysis buffer (50 mM MOPS, 15 mM $MgCl_2$, 150 mM NaCl, 100 $\mu g\, ml^{-1}$ cycloheximide, 0.5% Triton X-100, 1 $mg\, ml^{-1}$ Heparin, 0.2 $U\, \mu l^{-1}$ RNase inhibitor, 2 mM PMSF and 1 mM benzamidine) and centrifuged. After harvesting, soluble fraction was loaded onto the top of the pre-established sucrose gradient (10 ml of 10 ∼ 50%), and centrifuged at 36,000 r.p.m. in a Beckman SW-41 Ti rotor for 2 h at 4 °C. After ultracentrifugation, gradients were fractionated and collected using the ISCO tube piercer (Brandel) and fraction collector (Bio-Rad). Fractions were analysed by western blotting, where the signal intensities of each fraction were quantitated and analysed using the Multi Gauge software (version 3.0, Fujifilm).

**LC-MS/MS.** Immunoprecipitates of FLAG-CTIF-WT were analysed by SDS–PAGE and stained with silver nitrate. In-gel digestion and LC-MS/MS were conducted by ProteomTech, Korea.

**GST pull-down assay.** An *in vitro* GST pull-down assay was performed using recombinant proteins. Each recombinant protein was expressed in *Escherichia coli* BL21 (DE3) pLysS by adding 1 mM isopropylthiogalactoside when the OD at 600 nm reached 0.5. GST-tagged proteins and either His-eEF1A1 or His-DCTN were incubated in 800 μl of binding buffer (10 mM Tris-HCl (pH 8.0), 150 mM NaCl, 1% (vol/vol) glycerol, 0.001% BSA, 0.1% Triton X-100, 1 mM PMSF, 2 mM benzamidine and protease inhibitor cocktail tablets) at 4 °C for 1 h. After incubation, the mixture was added to Glutathione Sepharose 4B resin (Amersham-Pharmacia Biotech) and further incubated for 1 h. The mixture was washed five times with 1 ml of binding buffer. And then, the bead-bound proteins were resolved by SDS–PAGE and analysed by western blotting.

**TUNEL assay.** Terminal deoxynucleotidyl transferase dUTP nick end labelling (TUNEL) assay was performed using the *in situ* cell death detection kit (Roche) according to the manufacturer's instructions. Briefly, HeLa cells stably expressing CFTR-ΔF508 were transiently transfected with the indicated siRNA. Two days later, the cells were treated with MG132 for 12 h. The cells were fixed with 4% paraformaldehyde in PBS for 30 min at RT, and endogenous peroxidase activity was blocked using 3% hydrogen peroxide (Sigma) in PBS for 10 min at RT. The cells were permeabilized with 0.1% Triton X-100 in PBS for 10 min at RT. DNA strand breaks were then labelled with tetramethylrhodamine (TMR) red for 1 h at 37 °C. Nuclei were stained with DAPI for 5 min at RT. Cells were visualized using a Zeiss confocal microscope (LSM 510 Meta). To quantify the apoptotic cells, more than 100 cells were counted for each experiment. The results obtained from three independently performed TUNEL assays were counted and analysed.

For the complementation assay, HeLa cells stably expressing CFTR-ΔF508 were transiently transfected with the *CTIF* siRNA. The following day, the cells were re-transfected with siRNA-resistant pcDNA3-FLAG-$CTIF^R$-WT or 54–598.

**Immunohistochemistry.** Formalin-fixed, paraffin-embedded human cerebellum brain tissue samples of 5 μm thickness were obtained from Biochain. The PD patient was 73 years old, and an age-matched normal subject was used as a control. Brain samples were stained with haematoxylin and eosin to ensure the quality.

After deparaffinization, hydration and heat-induced antigen retrieval (10 mM sodium citrate, pH 6.0, microwave boiling, 3 min × 5 times), the sections were placed in 0.3% hydrogen peroxide in methanol for 10 min. After incubation in a blocking solution (5% BSA, 0.3 Triton X-100 in PBS (pH 7.4)) at RT for 1 h, the tissues were incubated with anti-α-synuclein antibody (1:50, BD Biosciences) and anti-CTIF antibody (1:50) at 4 °C for 72 h. For immunohistochemistry, sections were incubated in biotinylated secondary antibodies (α-rabbit IgG 1:500, α-mouse IgG 1:500, Vector Laboratories) at RT for 1 h, and then incubated with an ABC kit (Vector Laboratories) for 30 min. Sections were reacted with 3,3′-diaminobenzidine (DAB) (Vector Laboratories), and then dehydrated and coverslipped.

**Double-immunofluorescent histochemistry.** For double-immunofluorescent histochemistry, slides were incubated with Alexa Fluor 568 donkey α-mouse IgG secondary antibody (Life Technologies, 1:1,000) for the detection of α-synuclein and Alexa Fluor 488 donkey α-rabbit IgG (H + L) Antibody for the detection of CTIF at RT for 1 h. For visualization of the nuclei, sections were incubated with 0.2 $\mu l\, ml^{-1}$ 4′,6-diamidino-2-phenylindole (DAPI; Sigma) in PBS for 10 min. After washing, the sections were mounted in Vectashield (Vector Laboratories).

**Statistical analysis.** Throughout the paper, quantitations are shown as means ± s.d.'s. Statistical analyses were performed with two-tailed, equal-sample variance Student's *t*-tests. In all cases, statistical significance was considered with $P$ value < 0.05.

**Data availability.** The data that support the findings of this study are available from the corresponding author upon reasonable request.

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

## Acknowledgements
We thank Ron R. Kopito for providing plasmid expressing CFTR-ΔF508, Elizabeth S. Sztul for plasmid expressing GFP-250, Tso-Pang Yao for plasmid expressing FLAG-HDAC6,

Michael Y. Sherman for plasmids expressing GFP-SYN1 and mRFP-Ub, Eui-Ju Choi for plasmid expressing FLAG-SOD1, Berndt Müller for the antibody against human SLBP, Sung Hoon Back for MEF-eIF2α (A/A) cells and Sung Key Jang for plasmid expressing GST. This work was supported by the National Research Foundation of Korea (NRF) grant funded by the Korea Government (MSIP; NRF-2015R1A3A2033665).

## Author contributions

J.P. and Y.P. designed and performed experiments, analysed data and wrote the paper. I.R., K.M.K., J.C., M.-H.C., H.J.L., N.O. and K.K. designed and performed experiments and analysed data. C.L., J.-H.B. and Y.K.K. analysed data. Y.K.K. designed and supervised the study and wrote the manuscript.

## Additional information


**Competing interests:** The authors declare no competing financial interests.



