## [Peer review file · Nature Communications]

Reviewers' comments:

Reviewer #1 (Remarks to the Author):

The authors present very intriguing and novel observation that the translation factor CTIF plays a major role in formation of aggresome. This finding represents a major advance in the field. Experiments are well done and the results are clear. However, many conclusions of the study are based on circumstantial evidences and require further direct tests.

1. They show that CED complex associates with aggresome. To test their model, it is critical to show that it interacts with aggregate precursors in the process of transport.
2. They need to show directly that CE is involved in transport. Just showing that depletion of components of CED leads to appearance of multiple aggregates in the cell is not enough.
3. Does CED interact with misfolded soluble proteins? Does it interact with DRiPs?
4. Does eEF1A recruit substrates to CED? Does CTIF link eEF1A with DCTN1?
5. What is the relation of CED and HDAC6? Are these parallel pathways of transport of aggregates? How about Bag3-dependent transport?

Reviewer #2 (Remarks to the Author):

In this report, Park et. al. describes the identification of a ribosome-mRNA adaptor protein, CTIF, as a factor that connects nascent misfolded proteins to dynactin, and thereby enables the transport of misfolded proteins by dynein motors to form the perinuclear inclusions, the aggresomes. Growing evidence has indicated that failure to degrade nascent misfolded proteins by proteasomes can drive aggresome formation. eEF1A1, a translational regulator that binds and chaperones misfolded nascent polypeptides towards 26S proteasome for degradation, was recently shown to activate aggresome formation when proteasome activity is inhibited. The current report extends the previous study and identified CTIF as an eEF1A1- and dynactin-interactive protein, implicating a role of CTIF in eEF1A1- and dynein-mediated misfolded protein concentration and aggresome formation. Biochemical data were presented to indicate that MG132 treatment dissociated CTIF from components involved in translation and mRNA while the same treatment did not affect CTIF-dynactin or CTIF-eEF1A1 interaction- evidence that proteasome inhibition biased CTIF interaction towards dynein motors. Lastly, the authors showed that CTIF knockdown suppressed aggresome formation. Overall, the evidence presented in the report is in support of the conclusion that CTIF is required for aggresome formation; however, additional data are needed to strengthen the "adaptor" model and justify the publication in Nature Communication.

Specific Comments:

1. To support an active role in "concentrating" misfolded proteins, it is necessary to determine whether endogenous CTIF is concentrated at the protein aggregates prior to their accumulation at the perinuclear region. The presence or absence of dynein motors and eEF1A1 at the dispersed aggregates in CTIF knockdown cells should be determined. Conversely, the relationship between CTIF and protein aggregates in eEF1A1 knockdown cells should be assessed. To exclude the possibility that CTIF knockdown prevents misfolded protein ubiquitination, the status of ubiquitination should also be examined.
2. The use of ectopically expressed CTIF in Figure 1d and supplementary Fig S2 to document the presence of CTIF in aggresome is not ideal, as robust production of CTIF could, in itself, contribute to aggresome formation. A characterization of endogenous CTIF in aggresomes should be taken. In Fig 1e, where endogenous CTIF was examined, there was only one cell presented and the staining of vimentin was also atypical. A better documentation of endogenous CTIF in aggresomes should be presented.
3. Similar to the previous characterization of eEF1A1, it remains unclear how inhibition of

proteasome activity would activate CTIF-dependent misfolded protein transport. A brief discussion on various possibilities would be helpful.

Responses to reviewer #1's comments

The authors present very intriguing and novel observation that the translation factor CTIF plays a major role in formation of aggresome. This finding represents a major advance in the field. Experiments are well done and the results are clear. However, many conclusions of the study are based on circumstantial evidences and require further direct tests.

➤ **We appreciate the reviewer's evaluation of our work.**

1. They show that CED complex associates with aggresome. To test their model, it is critical to show that it interacts with aggregate precursors in the process of transport.

➤ **We agree with the reviewer. To address the reviewer's concern, we performed additional IP experiments using extracts of the cells treated with nocodazole (Supplementary Fig. 7 in the revised manuscript), which inhibited aggresome formation, and consequently triggered the accumulation of small cytoplasmic aggregates (Fig. 1d and Supplementary Fig. 2b in the revised manuscript). Under the conditions, we assessed the levels of co-immunoprecipitated polypeptidyl-puro in the IPs of eEF1A1 or CTIF. Polypeptidyl-puro typifies newly synthesized misfolded polypeptides and DRiPs because puromycin triggers premature termination of translation. The results showed that polypeptidyl-puro coimmunopurified with eEF1A1 or CTIF. Interestingly, the associations with polypeptidyl-puro were not significantly affected by treatment with nocodazole. These results indicate that the CED complex associates with misfolded polypeptides largely before aggresomal targeting and formation. The new data were added to Supplementary Fig. 7, and a description of the data was added to the Results section (pages 11–12) in the revised manuscript.**

2. They need to show directly that CE is involved in transport. Just showing that depletion of components

of CED leads to appearance of multiple aggregates in the cell is not enough.

- **We appreciate the reviewer's comment and agree with the reviewer. We think that a single-molecule approach is the best way to investigate a direct role of CTIF and eEF1A1 in aggresomal transport. However, because of technical limitations, it is very hard to carry out the experiment. A survey of the literature revealed that most studies on HADC6- or Bag3-mediated transport of misfolded polypeptides have employed similar approaches to those used in this study.**
- **Although we did not perform a single-molecule approach, our data in this study clearly demonstrated the critical role of CED in the transport of misfolded polypeptides toward aggresome: (i) CED components are colocalized with the known aggresomal polypeptides and marker proteins (Fig. 1 and Supplementary Fig. 3); (ii) downregulation of the CED component disrupts aggresome formation (Fig. 3 and Supplementary Fig. 6); (iii) the N-terminal region spanning amino acids 12–53 of CTIF is critical and sufficient for aggresomal targeting (Fig. 4 and Supplementary Fig. 9); (iv) an interaction between the CED complex and misfolded polypeptides occurs largely before aggresomal targeting and formation (Supplementary Fig. 7); and (v) endogenous CTIF is concentrated at the cytoplasmic aggregates containing misfolded polypeptides before aggresomal targeting and formation. Therefore, our data clearly support an essential role of the CED complex in the transport of misfolded polypeptide.**
- **We are currently collaborating with other group to determine a direct role of CTIF and eEF1A1 in the transport of misfolded polypeptides using a single-molecule approach. Therefore, we would prefer to conduct this investigation in a separate future study.**

3. Does CED interact with misfolded soluble proteins? Does it interact with DRiPs?

- **Please refer to our responses to reviewer #1's comment #1. We employed a nocodazole treatment for an accumulation of small cytoplasmic aggregates containing misfolded polypeptides. We then performed IPs of eEF1A1 and CTIF to assess the amount of co-**

immunopurified polypeptidyl-puro, which typifies DRiPs (Lelouard, et al., 2004; Szeto, et al., 2006). The results indicate that the CED complex associated with misfolded polypeptides largely before aggresomal targeting and formation. The new data were added to Supplementary Fig. 7, and a description of the data was added to the Results section (pages 11-12) in the revised manuscript.

4. Does eEF1A recruit substrates to CED? Does CTIF link eEF1A with DCTN1?

- We appreciate the reviewer's keen comments. As the reviewer pointed out, it is critical to determine whether CTIF links eEF1A1 and DCTN1. To this end, we performed IPs of FLAG-eEF1A1 using extracts of the cells depleted of endogenous CTIF. The results revealed that downregulation of CTIF disrupted the association between eEF1A1 and DCTN1. These data support our conclusion that CTIF bridges eEF1A1 and DCTN1. The new data and description were added to Supplementary Fig. 5d,e and the Results section (pages 9-10) in the revised manuscript, respectively.
- To investigate whether eEF1A1 recruits substrates to CED, we performed additional IPs using extracts of the cells treated with nocodazole. The results showed that the association between CTIF or eEF1A1 with polypeptidyl-puro was not significantly affected by treatment with nocodazole (Supplementary Fig. 7b,c), indicating that the CED complex associates with misfolded polypeptides largely before aggresomal targeting and formation. Interestingly, downregulation of eEF1A1 abolished the association between CTIF and polypeptidyl-puro (Supplementary Fig. 7d,e), indicating that the recruitment of misfolded-polypeptides to CED complex is dependent on eEF1A1. The new data were added to Supplementary Fig. 7d,e, and a description of the data was added to the Results section (pages 11-12) in the revised manuscript.

5. What is the relation of CED and HDAC6? Are these parallel pathways of transport of aggregates? How

about Bag3-dependent transport?

- **In response, multiple lines of evidence in this study support the cooperative role of the CED and HDAC6 in the formation of polyubiquitin-enriched aggresomes. The relationship between CED and HDAC6 was described in the Discussion section in the revised manuscript (pages 19–20).**
- **CED-, HDAC6-, and Bag3-mediated aggresome formation commonly induces the sequestration of ubiquitinated misfolded polypeptides into aggresomes. Therefore, it is likely that Bag3 may function in cooperation with CED and HDAC6. Alternatively, these mechanisms may act in a complementary manner sharing a common factor, the dynein motor. We have added a brief description of the relationship to the Discussion section in the revised manuscript (pages 20–21).**

Responses to reviewer #2's comments

In this report, Park et. al. describes the identification of a ribosome-mRNA adaptor protein, CTIF, as a factor that connects nascent misfolded proteins to dynactin, and thereby enables the transport of misfolded proteins by dynein motors to form the perinuclear inclusions, the aggresomes. Growing evidence has indicated that failure to degrade nascent misfolded proteins by proteasomes can drive aggresome formation. eEF1A1, a translational regulator that binds and chaperones misfolded nascent polypeptides towards 26S proteasome for degradation, was recently shown to activate aggresome formation when proteasome activity is inhibited. The current report extends the previous study and identified CTIF as an eEF1A1- and dynactin-interactive protein, implicating a role of CTIF in eEF1A1- and dynein-mediated misfolded protein concentration and aggresome formation. Biochemical data were presented to indicate that MG132 treatment dissociated CTIF from components involved in translation and mRNA while the same treatment did not affect CTIF-dynactin or CTIF-eEF1A1 interaction- evidence that proteasome inhibition biased CTIF interaction towards dynein motors. Lastly, the authors showed that CTIF knockdown suppressed aggresome formation. Overall, the evidence presented in the report is in support of the conclusion that CTIF is required for aggresome formation; however, additional data are needed to strengthen the “adaptor” model and justify the publication in Nature Communication.

➤ **We appreciate the reviewer's evaluation of our work.**

Specific Comments:

1. To support an active role in “concentrating” misfolded proteins, it is necessary to determine whether endogenous CTIF is concentrated at the protein aggregates prior to their accumulation at the perinuclear region. The presence or absence of dynein motors and eEF1A1 at the dispersed aggregates in CTIF knockdown cells should be determined. Conversely, the relationship between CTIF and protein aggregates in eEF1A1 knockdown cells should be assessed. To exclude the possibility that CTIF

knockdown prevents misfolded protein ubiquitination, the status of ubiquitination should also be examined.

- **We agree with the reviewer. Reviewer #1 also raised the same concern (Reviewer #1's first comment): "They show that CED complex associates with aggresome. To test their model, it is critical to show that it interacts with aggregate precursors in the process of transport". To address the reviewer's concern, we performed additional IP experiments using extracts of the cells treated with nocodazole (Supplementary Fig. 7 in the revised manuscript), which inhibited aggresome formation, and consequently triggered the accumulation of small cytoplasmic aggregates (Fig. 1d and Supplementary Fig. 2b in the revised manuscript). Under such conditions, we assessed the levels of co-immunoprecipitated polypeptidyl-puro in the IPs of eEF1A1 or CTIF. Polypeptidyl-puro typifies newly synthesized misfolded polypeptides and DRiPs because puromycin triggers premature termination of translation. The results showed that polypeptidyl-puro coimmunopurified with eEF1A1 or CTIF. Interestingly, the associations with polypeptidyl-puro were not significantly affected by treatment with nocodazole. These results indicate that the CED complex associates with misfolded polypeptides largely before aggresomal targeting and formation. The new data were added to Supplementary Fig. 7, and a description of the data was added to the Results section (pages 11–12) in the revised manuscript.**
- **To address the reviewer's second concern, we performed additional immunostainings. The results showed that each CED component overlapped with aggresomes containing polypeptidyl-puro when cells were treated with MG132. Importantly, eEF1A1 and DCTN1 also overlapped with small cytoplasmic aggregates of polypeptidyl-puro accumulated by CTIF downregulation. However, CTIF was partially colocalized with small cytoplasmic aggregates of polypeptidyl-puro accumulated by eEF1A1 downregulation, because CTIF itself lacks the ability to bind to misfolded polypeptides. These results suggest that CTIF is required for the efficient transport of misfolded polypeptides associated with eEF1A1. The new data were added to Supplementary Fig. 8, and a description of the data was added to the Results section**

(page 12) in the revised manuscript. A possible explanation for why DCTN1 overlapped with small cytoplasmic aggregates of polypeptidyl-puro accumulated by CTIF downregulation was added to the Discussion section in the revised manuscript (page 21).

- **As pointed out by the reviewer, we also determined whether CTIF knockdown prevents misfolded protein ubiquitination. The results revealed that CTIF downregulation did not significantly affect K63-ubiquitination of endogenous proteins. The new data were added to Supplementary Fig. 6e in the revised manuscript.**

2. The use of ectopically expressed CTIF in Figure 1d and supplementary Fig S2 to document the presence of CTIF in aggresome is not ideal, as robust production of CTIF could, in itself, contribute to aggresome formation. A characterization of endogenous CTIF in aggresomes should be taken. In Fig 1e, where endogenous CTIF was examined, there was only one cell presented and the staining of vimentin was also atypical. A better documentation of endogenous CTIF in aggresomes should be presented.

- **As pointed out by the reviewer, we repeated the immunostaining experiments using endogenous CTIF. All new data using endogenous CTIF showed the same pattern observed in immunostainings using FLAG-CTIF. Therefore, the original data (Fig. 1d in the original manuscript) were replaced by new data (Fig. 1d in the revised manuscript), and the original Fig. 1d was moved to Supplementary Fig. 2b in the revised manuscript.**
- **For Fig. 1e, we also repeated the same immunostaining experiments using endogenous CTIF and vimentin. The new data showed that the CTIF-containing aggresomes were surrounded by vimentin. Therefore, the original data (Fig. 1e in the original manuscript) were replaced by new data (Fig. 1e in the revised manuscript).**

3. Similar to the previous characterization of eEF1A1, it remains unclear how inhibition of proteasome activity would activate CTIF-dependent misfolded protein transport. A brief discussion on various possibilities would be helpful.

- **Based on recent structural and single-molecule experiments, we have provided a plausible possibility in the Discussion section (pages 18–19 in the revised manuscript).**

Reviewers' comments:

Reviewer #1 (Remarks to the Author):

The authors reasonably addressed almost all the questions raised in the previous review cycle. However, the question about the relationship between the CTIF-eEF1A pathway and Bag3 and HDAC6 pathways remains unaddressed. Are these parallel pathways of aggregate transport or they physically cooperate? This question needs to be resolved.

Reviewer #2 (Remarks to the Author):

The revision has addressed all key comments.

Responses to reviewer #1's comments

The authors reasonably addressed almost all the questions raised in the previous review cycle. However, the question about the relationship between the CTIF-eEF1A pathway and Bag3 and HDAC6 pathways remains unaddressed. Are these parallel pathways of aggregate transport or they physically cooperate? This question needs to be resolved.

- **We appreciate the reviewer's comments and agree with the reviewer. To address the reviewer's concern, we performed additional IP experiments using extracts of the cells treated or not treated with MG132. The data showed that MG132 treatment promoted the association between endogenous BAG3 and CTIF. We also performed confocal microscopy to monitor a change in the distribution of the polyubiquitin-enriched aggresome containing misfolded CFTR-ΔF508. The results showed that BAG3 downregulation triggered the dispersion of aggresome containing CFTR-ΔF508 into small cytoplasmic aggregates. These data indicate that the CED complex functions cooperatively with BAG3 to trigger the aggresome formation of misfolded and polyubiquitinated proteins. The new data were added to Supplementary Fig. 14, and a description of the data was added to the Discussion section (page 20) in the revised manuscript.**
- **Although we do not know the underlying molecular details, the new data provide sufficient evidence for the cooperative action of CED, HDAC6, and BAG3. We would prefer to conduct the further investigation for the molecular details in a separate future study.**

Responses to reviewer #2's comments

The revision has addressed all key comments.

- **We appreciate the reviewer's evaluation of our work.**

REVIEWERS' COMMENTS:

Reviewer #1 (Remarks to the Author):

Authors reasonably addressed my prior questions about about HDAC6 and Bag3. However they forgot to modify their model on Fig. 7c accordingly.

Responses to reviewer #1's comment

Authors reasonably addressed my prior questions about about HDAC6 and Bag3. However they forgot to modify their model on Fig. 7c accordingly.

- **We appreciate the reviewer's comment. The reviewer suggested us including Bag3 in our model. However, our manuscript focuses on the findings on a new functional complex involved in aggresome formation of misfolded polypeptides. To emphasize our new observations, we would like to keep the model as it stands.**

- **In the current version of manuscript, we did not intensively investigate the relationship between CED, BAG3, and HDAC6. Therefore, we would like to add BAG3 and HDAC6 to our model after future intensive work.**